# Extracellular adenosine deamination primes tip organizer development in *Dictyostelium*

Pavani Hathi, Baskar Ramamurthy*

Bhupat and Jyoti Mehta School of Biosciences, Department of Biotechnology, Indian Institute of Technology-Madras, Chennai, India

**\*For correspondence:**
rbaskar@iitm.ac.in

**Competing interest:** The authors declare that no competing interests exist.

## eLife Assessment

During the development of the unicellular eukaryote Dictyostelium discoideum, cells aggregate into mounds, forming protrusions or tips, which then become the front of migrating slugs and the top of fruiting bodies. This **valuable** study identifies adenosine deaminase-related growth factor (ADGF) as a key regulator of tip formation and **convincingly** shows that ADGF catalyses the conversion of adenosine to ammonia, allowing ammonia to initiate tip formation, and then elucidates pathways upstream and downstream of ADGF. The authors discuss the intriguing possibility that mammalian ADGF may also similarly regulate development.

**Abstract** Ammonia is a morphogen in *Dictyostelium* and is known to arise from the catabolism of proteins and RNA. However, we show that extracellular adenosine deamination catalysed by 'adenosine deaminase-related growth factor' (ADGF) is a major source of ammonia and demonstrate a direct role of ammonia in tip organizer development. The tip formed during early development in *Dictyostelium* functions analogously to the embryonic organizer of higher vertebrates. *Dictyostelium* strains carrying mutations in the gene *adgf* fail to establish an organizer, and this could be reversed by exposing the mutants to volatile ammonia. Interestingly, *Klebsiella pneumoniae* physically separated from the *Dictyostelium adgf* mutants in a partitioned dish also rescues the mound arrest phenotype. Both the substrate, adenosine and the product, ammonia, regulate *adgf* expression, and ADGF acts downstream of the histidine kinase DhkD in regulating tip formation. Thus, the consecutive transformation of extracellular cAMP to adenosine, and adenosine to ammonia is integral steps during *Dictyostelium* development. Remarkably, in higher vertebrates, *adgf* expression is elevated during gastrulation and thus adenosine deamination may be a conserved process driving organizer development in different organisms.

## Introduction

During early embryonic development, organizers play an important role in patterning and directing the differentiation of surrounding cells into specific tissues and organs. The embryonic organizer establishes the developmental polarity in vertebrates; and similarly, in *Dictyostelium,* the mound/slug tip acts as an organizer (***Rubin and Robertson, 1975***) playing a pivotal role in guiding collective cell migration and patterning. Despite numerous investigations on *Dictyostelium* tip formation, the processes by which the tip establishes and maintains the primary developmental axis remain elusive. Gaining insights into the mechanisms regulating tip organizer function will offer valuable insights on orchestrated cell movements and processes underlying development.

*Dictyostelium* transitions from a unicellular amoeba to a multicellular organism in response to starvation. The cells aggregate into a mound, which gives rise to a migrating slug and ultimately a fruiting body composed of spores and a dead stalk (*Raper, 1940*; *Kessin, 2001*). This process is regulated by diffusible signals including cAMP, adenosine, ammonia, and a chlorinated hexaphenone, 'differentiation-inducing factor' (DIF) (*Bloom and Kay, 1988*; *Williams, 1988*; *Gross, 1994*; *Mahadeo and Parent, 2006*). Importantly, the cell cycle phase soon after starvation is known to strongly influence the cell fate either as prestalk (pst) or prespore (psp) cells (*Weeks and Weijer, 1994*; *Jang and Gomer, 2011*).

Tip development in *Dictyostelium* depends on cAMP signalling in turn regulating protein kinase A (PKA) activity, adenosine signalling and morphogenetic cell movements (*Schaap and Wang, 1986*; *Mann and Firtel, 1993*; *Siegert and Weijer, 1995*). Adenosine, a by-product of cAMP hydrolysis, acts as an inhibitory morphogen suppressing additional tip formation (*Schaap and Wang, 1986*).

Adenosine deaminases (ADA) catalyse the breakdown of adenosine to generate inosine and ammonia and are conserved among bacteria, invertebrates, vertebrates including mammals (*Cristalli et al., 2001*). In humans, two isoforms of ADA are known, including ADA1 and ADA2, and the *Dictyostelium* homolog of ADA2 is adenosine deaminase-related growth factor (ADGF). Unlike ADA that is intracellular, ADGF is extracellular and also has a growth factor activity (*Li and Aksoy, 2000*; *Iijima et al., 2008*). Loss-of-function mutations in *ada2* are linked to lymphopenia, severe combined immunodeficiency (*Bobby Gaspar, 2010*), and vascular inflammation due to accumulation of toxic metabolites like dATP (*Zhou et al., 2014*; *Notarangelo, 2016*). In mice, *ada2* disruption leads to perinatal mortality (*Wakamiya et al., 1995*), and overexpression of *ada2* results in aberrant heart and kidney development (*Riazi et al., 2005*). In frogs, mutations in *ada2* manifest in reduced body size and altered polarity (*Iijima et al., 2008*). In *Drosophila*, certain isoforms of ADGF are known to play a pivotal role in cell proliferation by depleting extracellular adenosine (*Zurovec et al., 2002*), and loss of function of *adgfA* promotes melanotic tumour formation and larval death (*Dolezal et al., 2005*). Adenosine deaminases are known to interact with dipeptidyl peptidase IV (DPP), cluster of differentiation (CD26) expressed on T-cells and the adenosine receptor, A2AR (expressed on dendritic cells) to facilitate cell–cell signalling (*Moreno et al., 2018*). Thus, *adgf* plays a pivotal role in the regulation of cell proliferation and development in several organisms.

Four isoforms of ADA are annotated in the *Dictyostelium discoideum* genome (*Eichinger et al., 2005*) including adenosine deaminase (*ada*; DDB_G0287371), adenosine deaminase acting on tRNA-1 (DDB_G0278943), adenosine deaminase tRNA-specific (DDB_G0288099) and adenosine deaminase-related growth factor (*adgf*; DDB_G0275179), and their role in growth and development is not known. In this study, we demonstrate that ammonia generated by ADGF plays a direct role in establishing the tip organizer, highlighting a novel link between nucleoside metabolism and developmental polarity in *Dictyostelium*.

## Results

### Differential regulation of *adgf* expression during growth and development

*Dictyostelium* strains carrying mutations in the gene *adgf* were obtained from the genome wide *Dictyostelium* insertion (GWDI) bank, and were subjected to further analysis to know the role of *adgf* during *Dictyostelium* development. To verify the insertion of blasticidin (*bsr*) resistance cassette in *adgf* mutant, a diagnostic PCR was carried out and the integration was validated (*Figure 1A, B*), and quantitative real-time PCR (qRT-PCR) analysis confirmed the absence of *adgf* expression (*Figure 1C*). In vertebrates, adenosine deaminases are expressed in a tissue-specific manner to control growth and development. To determine whether *adgf* expression is differentially regulated during *Dictyostelium* development, qRT-PCR was performed using RNA isolated at 0, 8, 12, 16, 20, and 24 hr post starvation. *adgf* expression peaks at 16 hr (*Figure 1D*), implying an important role for *adgf* later in development. At this time point, the expression of the other three isoforms of ADA was not significantly different from wild-type (WT), suggesting that the loss of *adgf* (data not shown) function is not compensated by the other isoforms.

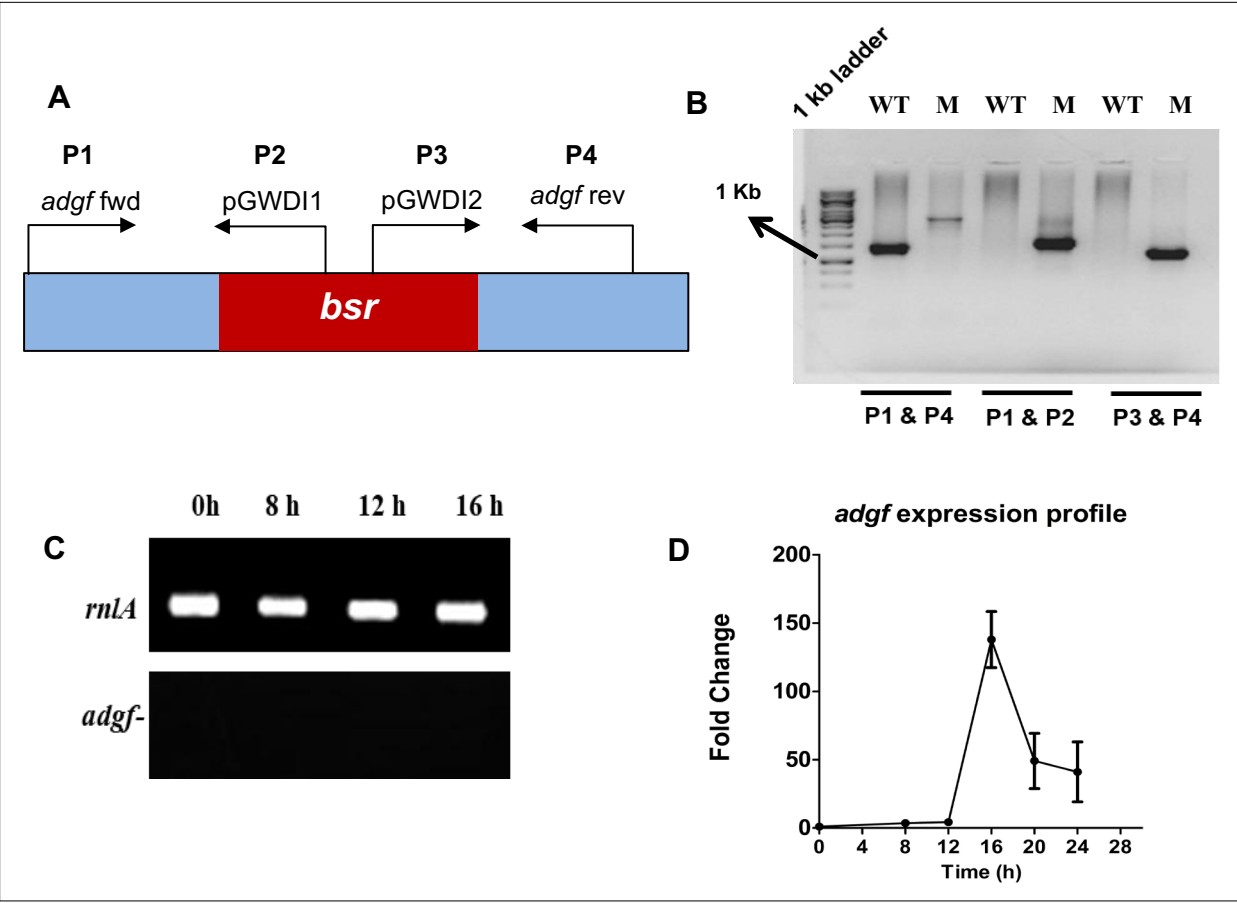

**Figure 1.** *adgf* mutant validation. (**A**) Schematic representation of *adgf* locus showing the relative positions of the primers (P1–P4) and the blasticidin resistance cassette (*bsr*). Primer P1 (*adgf* fwd) is at the start codon of *adgf*, and primer P4 (*adgf* rev) is 264 bp upstream of the stop codon, flanking the insertion site. Primers P2 (pGWD2) and P3 (pGWD1) are located within the *bsr* cassette. *bsr* insertion is in exon 2 of *adgf*. (**B**) PCR analyses using P1 and P4 primers. A 1.4-kb shift in the *adgf* mutant. PCR using P1 and P2 primers showed an amplicon from the mutant (M) and not from the wild-type (WT). PCR using P3 and P4 primers showed an amplicon with the *adgf* mutant while WT did not show any amplicon. (**C**) Semi-quantitative RT-PCR of the internal control, *rnlA* and *adgf*⁻. *adgf* expression during development in *Dictyostelium*. (**D**) Total RNA was isolated from *Dictyostelium* during vegetative growth and development using TRIzol method. To quantify *adgf* expression, qRT-PCR was carried out with *rnlA* as a control and the fold-change was calculated accordingly. Time points are shown in hours (bottom). Error bars represent the mean and SEM ($n = 3$).

The online version of this article includes the following source data and figure supplement(s) for figure 1:

**Source data 1.** PDF with original gel images for *Figure 1B, C*, showing the relevant bands.

**Source data 2.** Original files for gel images displayed in *Figure 1B, C*.

**Figure supplement 1.** Bioinformatic analyses of ADGF.

## The predicted structures of *Dictyostelium* ADGF and human ADA2 share a strong structural similarity

The *D. discoideum adgf* gene is predicted to encode a protein of 543 amino acids and belongs to the metallo-transferases superfamily (*Figure 1—figure supplement 1A*), having an ADA and an N-terminal domain similar to human ADA2 (*Figure 1—figure supplement 1B*). The ADGF sequence from the protein family database (https://www.ebi.ac.uk/interpro/) was analysed using the online tool SMART (Simple Modular Architecture Research Tool: http://smart.embl-heidelberg.de), to know the presence of structurally similar domains. The different domains of *Dictyostelium* ADGF (DdADGF) show a high degree of similarity with human ADA2. DdADGF shares 37.5% identity with human ADA2 and shares a sequence similarity of 59.5% with *D. fasciculatum*, 61.2% with *D. pupureum*, 53.6% with *D. lacteum* and 62.1% with *Polysphondylium pallidum* (https://www.uniprot.org/). Multiple sequence alignment reveals the presence of an N-terminal signal sequence characteristic of extracellular proteins

(*Figure 1—figure supplement 1C*) and conserved histidine and glutamine residues in the active site of both *D. discoideum* ADGF and human ADA2 (*Figure 1—figure supplement 1D*).

The phylogenetic relation of ADGF to the classic ADA subfamily has been reported previously (*Maier et al., 2005*). To determine the evolutionary relationship of DdADGF with that of other organisms, a phylogenetic analysis was carried out, and the amino acid alignment was created using MEGAX software (*Kumar et al., 2018*). The evolutionary history of ADGF was inferred using a maximum likelihood approach with bootstrap analysis (100 iterations) as described by *Felsenstein, 1985*. The resulting phylogenetic tree indicated that DdADGF is closely related to the ADGF proteins of other Dictyostelids, including *D. purpureum*, *Heterostelium pallidum*, and *Cavenderia fasciculata*. DdADGF forms a distinct clade, likely representing a distant relative of its vertebrate homolog (*Figure 1—figure supplement 1E*). Using the crystal structure of human ADA2, Cat eye syndrome critical region protein 1 (CECR1) as a template (PDB-3LGD), the structure of DdADGF was predicted by homology modelling (https://alphafold.ebi.ac.uk/) (*Figure 1—figure supplement 1F–H*). Alignment of DdADGF with human ADA2 yielded a root mean square deviation value of 1 Å, indicating strong structural similarity (*Eidhammer et al., 2000*; *Koehl, 2001*).

## *adgf* controls aggregate size in *Dictyostelium*

To understand the role of *adgf* during *D. discoideum* development, *adgf* mutants were plated at a density of $5 \times 10^5$ cells/cm$^2$ on non-nutrient phosphate buffered agar plates and monitored thereafter. In comparison to WT, the aggregates of *adgf* $^-$ were larger, and thus the number of aggregates was fewer (*Figure 2A*) than the WT. To determine the pathways impairing the tissue size in the mutant, RNA expression of countin (*ctn*) and small aggregates (*smlA*) was examined, and their levels were reduced significantly compared to controls (*Figure 2B*). Counting factor regulates group size by reducing the expression of cell–cell adhesion proteins cadherin (*cadA*) and contact site A (*csaA*) (*Siu et al., 1985*; *Coates and Harwood, 2001*). Compared to WT, both cell-to-cell adhesion and *cadA*, *csaA* expression were higher in the *adgf* mutant (*Figure 2C, D*), suggesting that *adgf* regulates the overall size of the aggregates.

cAMP chemotaxis significantly influences cell–cell adhesion (*Konijn et al., 1967*), and to determine if chemotaxis is impaired in the *adgf* $^-$ lines, an under-agarose chemotaxis assay was carried out. The chemotactic activity was not significantly different between the two cell types (*Figure 2E*), suggesting that the increased mound size in the mutant is not due to altered chemotaxis.

## *adgf* mutants form large, tipless mounds

Subsequent to plating on KK2 agar, WT cells formed mounds by 8–9 hr and culminated to form fruiting bodies by the end of 24 hr. In contrast, the *adgf* $^-$ lines were blocked as rotating mounds with no tips till 30 hr (*Figure 2F*; *Figure 2—videos 1 and 2*). Such a mound arrest phenotype could be mimicked by adding the ADA specific inhibitor deoxycoformycin (DCF) to WT cells (*Figure 2G*). After 36 hr, a fraction of *adgf* $^-$ mounds formed fruiting bodies with bulkier spore sac and stalk (*Figure 2H, I*). This late recovery from the mound arrest may be due to the expression of other ADA isoforms after 36 hr.

## Reduced ADA activity leads to high adenosine levels in the *adgf* mutant

If *adgf* function is compromised, ADA enzyme activity is expected to be low. To verify this, total ADA activity in WT and *adgf* $^-$ mounds was measured using a commercial assay kit. This assay relies on the conversion of inosine (derived from ADA activity) to uric acid and was measured spectrophotometrically. Although ADA activity dropped significantly in the mutant (*Figure 3A*), it was not completely abolished, and possibly, other isoforms may have a basal activity both at 12 and 16 hr.

ADGF quenches extracellular adenosine, and if blocked, as expected in *adgf*$^-$, the mutants will have increased extracellular adenosine. Using a commercial kit, the total adenosine levels were measured, and both at 12 and 16 hr, the levels were elevated (*Figure 3B*), and this difference was highly significant at 16 hr. *adgf* expression at this time point was also high in WT cells. The elevated adenosine levels at 16 hr may result from reduced ADA activity and increased expression of genes such as 5′ nucleotidase (*5′nt*) and phosphodiesterases (*pdsA* and *regA*), all involved in adenosine formation (*Headrick and Willis, 1989*; *Gödecke, 2008*). Hence, their expression levels were examined. Although the expression of both *5′nt* and *regA* was enhanced at 8 hr, *pdsA* levels remained

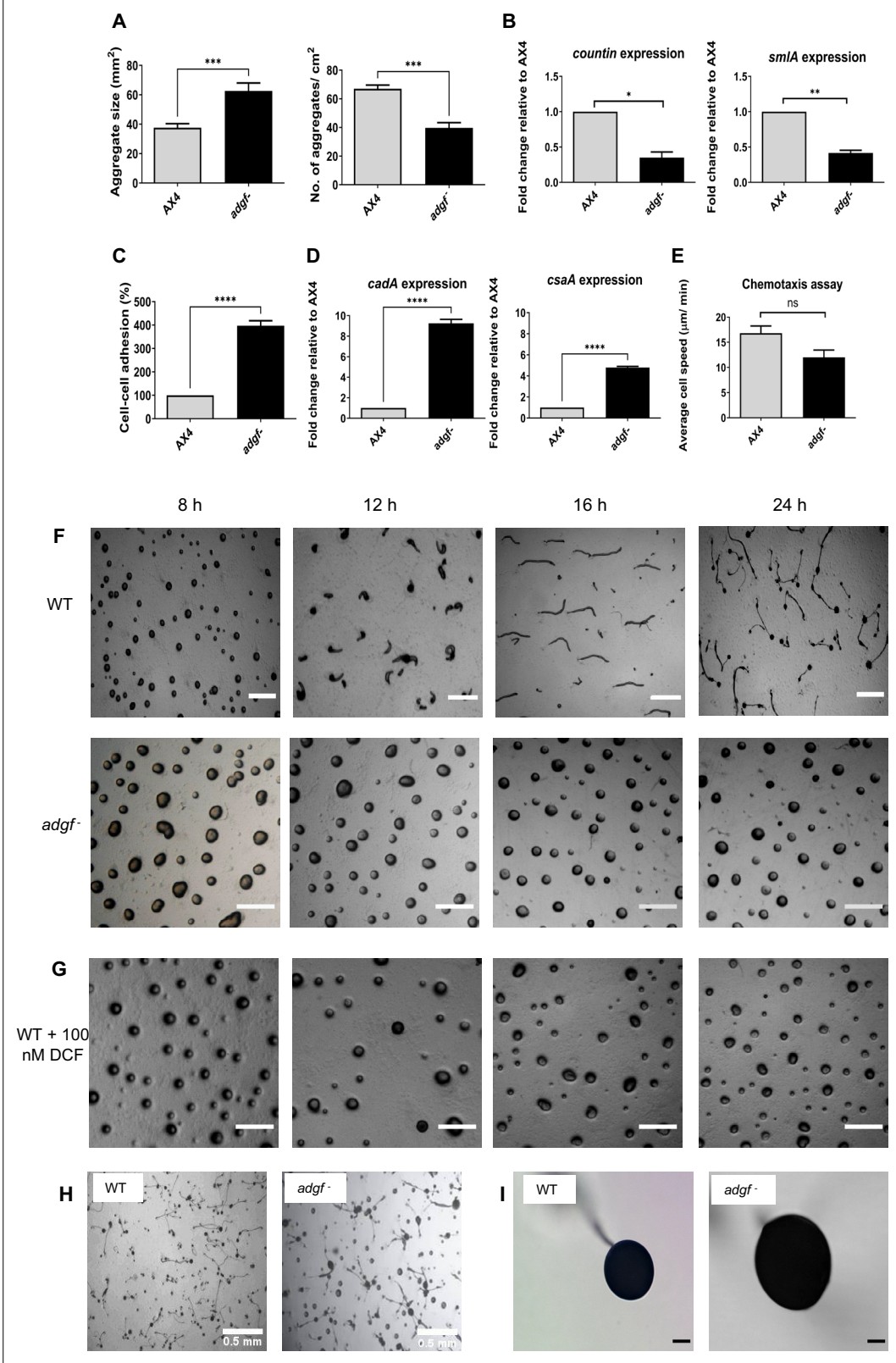

**Figure 2.** Aggregates formed by *adgf* mutants were larger in size. (**A**) The graph shows the mound size and the number of aggregates formed by WT and *adgf⁻*. A minimum of 20 aggregates were analyzed per experiment. The values represent mean ± SEM; *n* = 3 independent biological replicates. Significance level is indicated as \*\*\*p< 0.001 (Student's t-test). (**B**) Expression levels of the genes, countin (*ctn*) and small aggregates (*smlA*) during

*Figure 2 continued on next page*

*Figure 2 continued*

aggregation in *adgf⁻* compared to WT. *rnlA* was used as the internal control. Data represent mean ± SEM (n=3). Significance level is indicated as *p< 0.05, **p< 0.01 (Student's t-test). (**C**) WT and *adgf⁻* cells were developed on KK2 agar, and after 16 hr, the multicellular mounds/slugs were dissociated by vigorous vortexing in KK2 buffer. Individual cells were counted using a hemocytometer and resuspended in a phosphate buffer. Non-adherent single cells were counted 45 min after incubation. The percent cell–cell adhesion was plotted by normalizing the values to the non-adherent WT count to 100%. Error bars represent the mean ± SEM (*n* = 3). The level of significance is ****p< 0.0001 by Student's t-test. (**D**) qRT-PCR analysis of cadherin (*cadA*) and contact site (*csA*) during aggregation. The fold-change in RNA transcript levels is relative to WT at the indicated time points. Error bar is mean and SEM (*n* = 3). Significance level is indicated as ****p < 0.0001 (Student's *t*-test). (**E**) Under agarose chemotaxis assay. The average cell speed in response to 10 µM cAMP was recorded. A minimum of 25 cells were tracked for each experiment. The graph represents the mean ± SEM (*n* = 3). ns, non significant, by Student's t-test. Developmental phenotype of *adgf⁻*. (**F**) WT and *adgf⁻* cells were washed, plated on 1% KK2 agar plates at a density of $5 \times 10^5$ cells/cm², incubated in a dark, moist chamber and images were taken at different time intervals. (**G**) WT cells treated with 100 nM of DCF mimicked the mound arrest phenotype of the mutant. The time points are indicated in hours at the top of the figure. Scale bar: 2 mm (*n* = 3). (**H**) WT and *adgf⁻* cells after 36 hr of development. Scale bar: 0.5 mm (n=3). (**I**) Fruiting bodies of WT and *adgf⁻*. Scale bar: 2 mm. Atleast 30 fruiting bodies were analyzed for each experiment, *n* = 3 independent biological replicates.

The online version of this article includes the following video(s) for figure 2:

**Figure 2—video 1.** Timelapse video of wild-type AX4 development.

https://elifesciences.org/articles/104855/figures#fig2video1

**Figure 2—video 2.** Timelapse video of *adgf⁻* development.

https://elifesciences.org/articles/104855/figures#fig2video2

---

unaltered. Interestingly, the expression of all three genes trended lower at 12 hr but was significantly upregulated at 16 hr (*Figure 3C*), suggesting an important role of these genes at a specific time in development.

## Addition of ADA or overexpression of *adgf* cDNA restored tip development in *adgf⁻*

*adgf* mutants, which carry significantly high levels of adenosine. By administering the ADA enzyme on top of the *adgf⁻* mounds (*Figure 4A*), excess adenosine can be quenched, resulting in ammonia formation, possibly rescuing the phenotype. Indeed, the addition of 10 U ADA onto mutant mounds restored tip development. Besides, overexpression of WT *adgf* cDNA (driven by actin15 promoter) complemented the developmental defects of *adgf* mutants (*Figure 4B and C*), confirming that the developmental block is the result of *adgf* gene disruption alone. However, *adgf* cDNA overexpression in WT cells does not result in any observable defects (*Figure 4D*). Taken together, these findings support an important function of *adgf* in tip development.

## WT or its conditioned media (CM) restored tip formation in *adgf⁻* mounds

To determine whether *adgf* is necessary for tip formation in a cell-autonomous manner, WT and the mutant cells were mixed in different proportions and plated for development. In a mix of 50% WT: 50% mutant, the mound defects of *adgf⁻* were rescued, but with 20% WT and 80% *adgf⁻*, the rescue was partial (31 ± 4%) (*Figure 4E*) suggesting that the mound arrest phenotype of the mutant is due to the absence of some secreted factor(s).

Further, when developed in the presence of WT CM, *adgf⁻* cells formed tipped mounds and eventually fruiting bodies (*Figure 4F*). Conversely, in the presence of *adgf⁻* CM, WT cells developed as large mounds with no tips (*Figure 4G*) and the size was comparable to the mounds formed by *adgf⁻*. These findings imply that the developmental phenotype of *adgf⁻* is not due to a cell autonomous defect but due to faulty secreted factor signalling. While this finding is consistent with the involvement of a secreted factor, it is also possible that a membrane-bound extracellular factor may have a role in complementing the mutant phenotype.

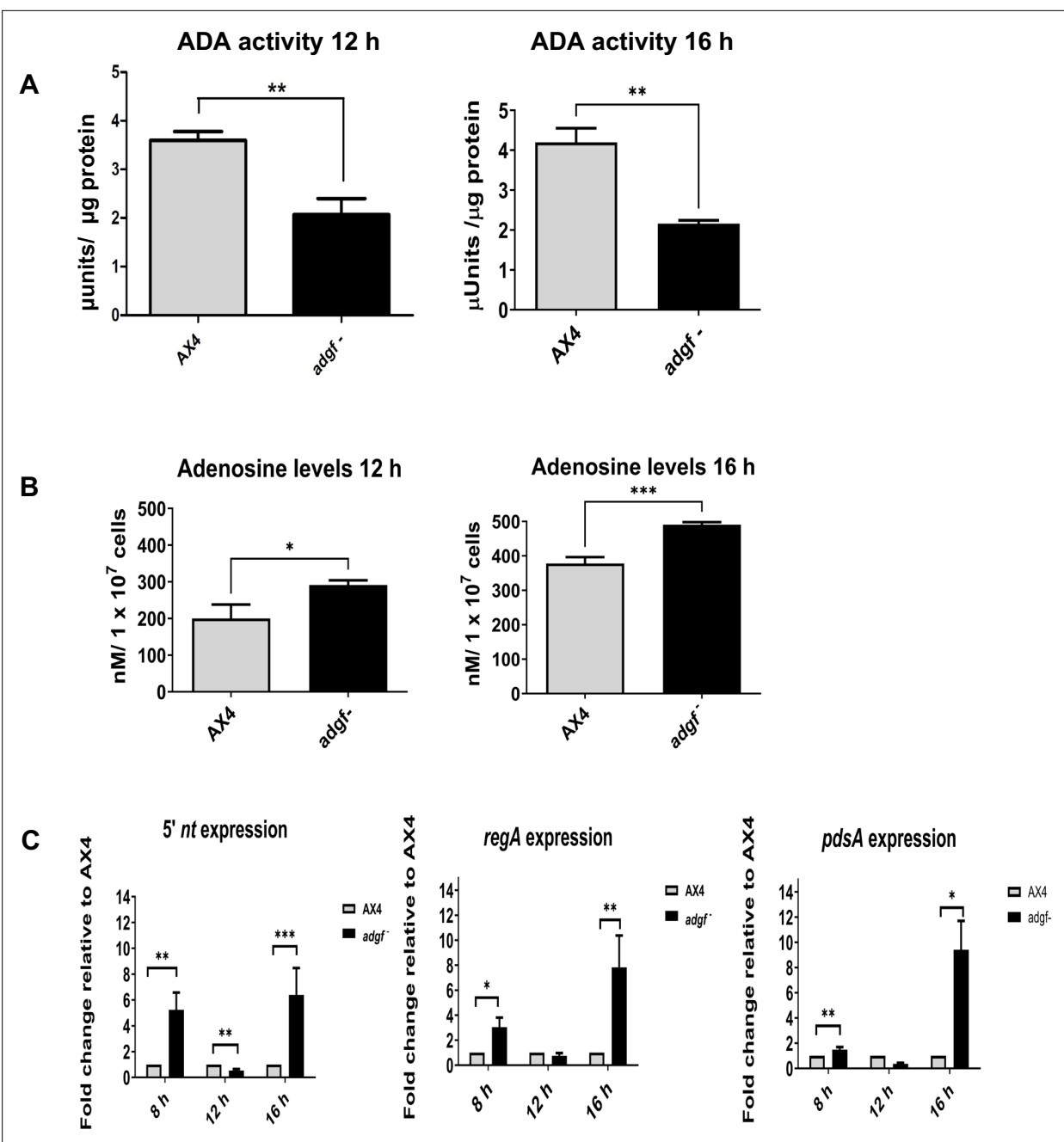

**Figure 3.** *adgf* mounds have reduced ADA activity and high adenosine levels. (**A**) ADA activity in WT and *adgf⁻* harvested at 12 and 16 hr. The enzymatic assay for ADA was performed in *adgf⁻* with the corresponding WT control. Error bars represent the mean and SEM (*n* = 3). Significance level is indicated as **p < 0.01 (Student's t-test). (**B**) Quantification of adenosine levels in WT and *adgf* mutants at 12 and 16 h. Level of significance is indicated as *p < 0.05, **p < 0.01, ***p < 0.001 by Student's t-test. Data represent mean and SEM (*n* = 3). (**C**) Expression profile of 5' nucleotidase (*5'nt*) and phosphodiesterases (*regA*, *pdsA*) involved in cAMP-to-adenosine conversion. The fold-change in RNA transcript levels is relative to WT at the indicated time points. *rnlA* was used as an internal control. Error bars represent the mean and SEM (*n* = 3). Level of significance is indicated as *p< 0.05, **p< 0.01, ***p< 0.001.

## Volatile ammonia rescued the mound arrest phenotype of *adgf⁻*

If ADGF enzyme activity is compromised, ammonia levels are expected to be low. Hence, the concentration of ammonia from WT and *adgf⁻* mounds was measured using a commercial kit. Ammonia levels were significantly reduced in *adgf⁻* lines (*Figure 5A*). By mixing sodium hydroxide and ammonium chloride (*Thadani et al., 1977*), ammonia could be generated, and in such conditions, tip formation

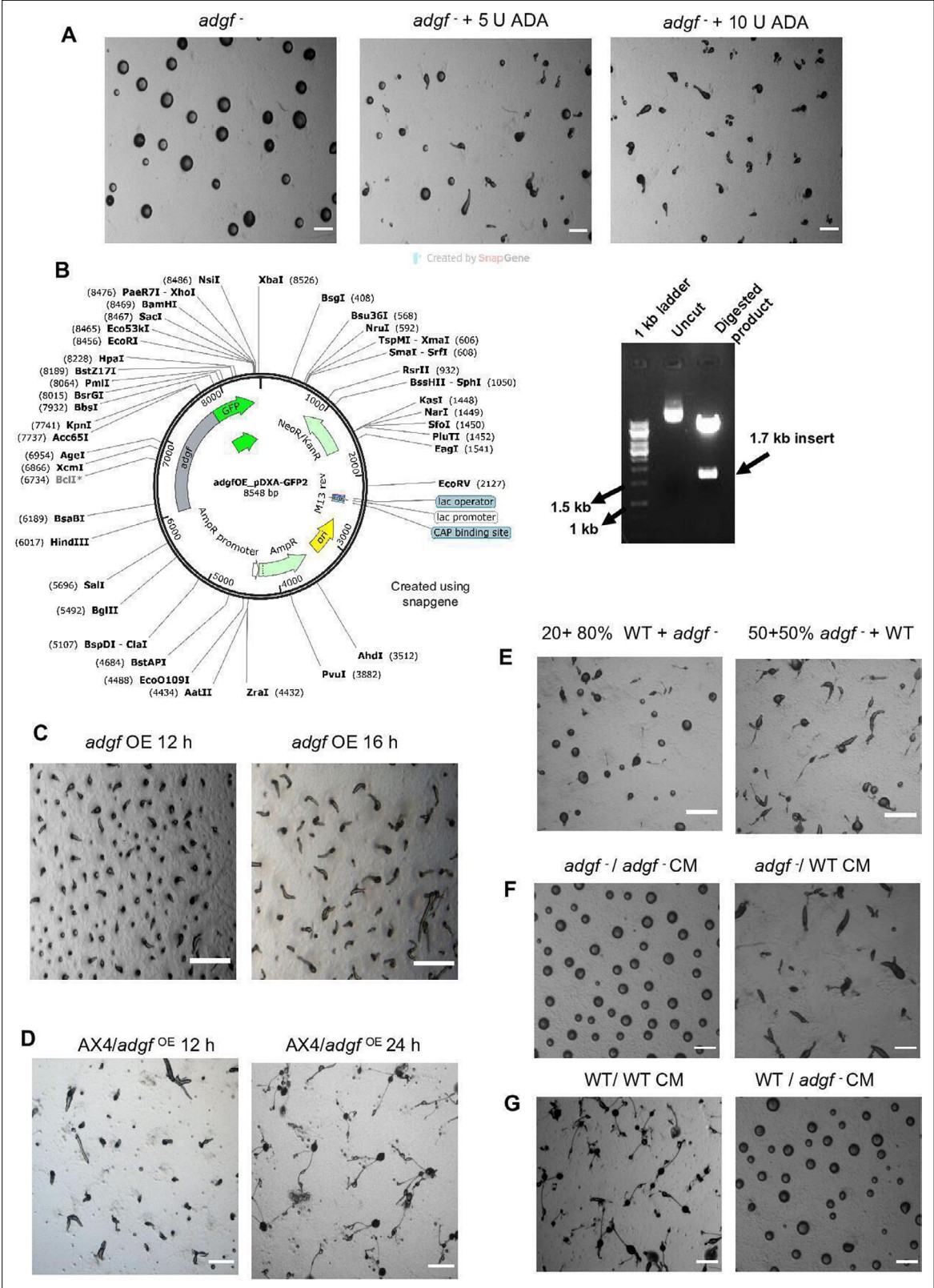

**Figure 4.** Overexpression of *adgf* rescued the mound arrest phenotype. (**A**) *adgf⁻* mounds were treated with 5 and 10 U ADA enzyme, and imaged at 16 hr. Scale bar: 2 mm (*n* = 3). (**B**) The full-length *adgf* gene was cloned in the vector pDXA-GFP2. The overexpression construct was verified by restriction digestion with HindIII and KpnI enzymes. (**C**) *adgf* overexpression in the mutant rescued the mound arrest. (**D**) Overexpression of *adgf* in WT background. Scale bar: 2 mm (*n* = 3). The time points in hours are shown at the top. WT cells mixed with *adgf⁻* rescued the *adgf* mutant phenotype. (**E**)

*Figure 4 continued on next page*

*Figure 4 continued*

Mixing of WT with *adgf⁻* in a 1:4 ratio showed a partial rescue, and a full rescue of the *adgf⁻* mound arrest phenotype in a 1:1 ratio with WT. Scale bar: 2 mm; (*n*=3). (**F**) Development of *adgf* mutants in the presence of *adgf⁻* CM and WT CM on KK2 agar plates. WT CM rescued the mound arrest. Scale bar: 2 mm; (*n*=3). (**G**) Development of WT in the presence of WT CM and *adgf⁻* CM on KK2 agar plates. *adgf⁻* CM induced mound arrest in WT cells. Scale bar: 2 mm (*n* = 3).

The online version of this article includes the following source data for figure 4:

**Source data 1.** PDF with original gel images for *Figure 4B*, showing the relevant bands.

**Source data 2.** Original files for gel images displayed in *Figure 4B*.

was restored in *adgf⁻* mounds (*Figure 5B*). All the rescue experiments involving ammonia were carried out for a 3.5-hr time period.

## Physically separated WT restored tip development in the mutant

WT mounds are expected to release a different set of volatiles including ammonia, possibly rescuing the mound arrest phenotype of the mutants nearby. To verify if this is correct, *adgf⁻* cells were developed in one half of a compartmentalized Petri dish, and WT cells on the other side. Interestingly, the mound arrest of *adgf⁻* was fully rescued (*Figure 5C*) in the presence of WT, although they were physically separated from each other. This suggests that volatiles, likely to be ammonia released from WT, are sufficient enough to rescue the mutant phenotype.

The *adgf⁻* CM is expected to have high adenosine levels, possibly generating low or no ammonia. Addition of ADA to *adgf⁻* CM in one compartment of the partitioned dish led to partial rescue (57% ± 2) of *adgf⁻* kept on the other side of the dish (data not shown), implying that the ammonia generated in such conditions may not be enough for a full rescue.

## Adenosine deamination alone drives the rescue of *adgf⁻*

To know if adenosine deamination is exclusively responsible for the rescue of the mutant, 10 ml cold KK2 buffer mixed with different concentrations of adenosine (10 µM, 0.1 mM, and 1 mM) was added in one half of the compartmentalized dish, and the other side of the dish had the mutant on phosphate buffered agar arrested at the mound stage. Addition of ADA enzyme (10 U) to the buffered solution containing adenosine led to a full rescue after 3.5 hr (*Figure 5D*), strongly indicating that volatile ammonia generated from adenosine deamination alone is rescuing the defect. Protein/RNA catabolism also generates ammonia in *Dictyostelium* (*White and Sussman, 1961*; *Hames and Ashworth, 1974*; *Schindler and Sussman, 1977*; *Walsh and Wright, 1978*), but their levels were not significantly different between WT and *adgf* mutants (*Figure 5—figure supplement 1*), suggesting that ammonia released from adenosine deamination plays a direct role in tip formation.

## *Klebsiella pneumoniae* in a partitioned dish rescues the *adgf⁻* mound arrest phenotype

Just like WT *Dictyostelium* rescuing the mound arrest phenotype of the *adgf* mutant, we examined if bacteria physically separated from the mutants would rescue the phenotype. To check this, *Klebsiella pneumoniae* and *adgf* mutants were incubated adjacent to each other within a compartmentalized KK2 agar plate. After a 12-hr incubation period, tip formation was restored in the mutants, while in the same time frame, the mounds in controls failed to form tips. Possibly, *Klebsiella* on KK2 plates with no nutrients would die, releasing ammonia and thereby restoring tip development in *Dictyostelium* (*Figure 5—figure supplement 2*).

## Caffeine, not ammonia, rescued the mound size of *adgf⁻*

Given that adenosine levels were elevated in the mutants, we attempted to rescue the large mound size observed in the *adgf* mutants by treating them with the adenosine receptor antagonist, caffeine (*Costenla et al., 2010*; *Ribeiro and Sebastião, 2010*). Caffeine rescued this early developmental defect in a dose-dependent manner (*Figure 5E*), suggesting that *adgf* may be one of the regulators of group size in *Dictyostelium*. Since ammonia levels were reduced in the mutant, we tested whether exposure to ammonia could rescue the large mound size of the *adgf* mutant. Exposing the mutants to 0.01 M ammonia soon after plating or 6 hr after plating had no effect on the mound size of the

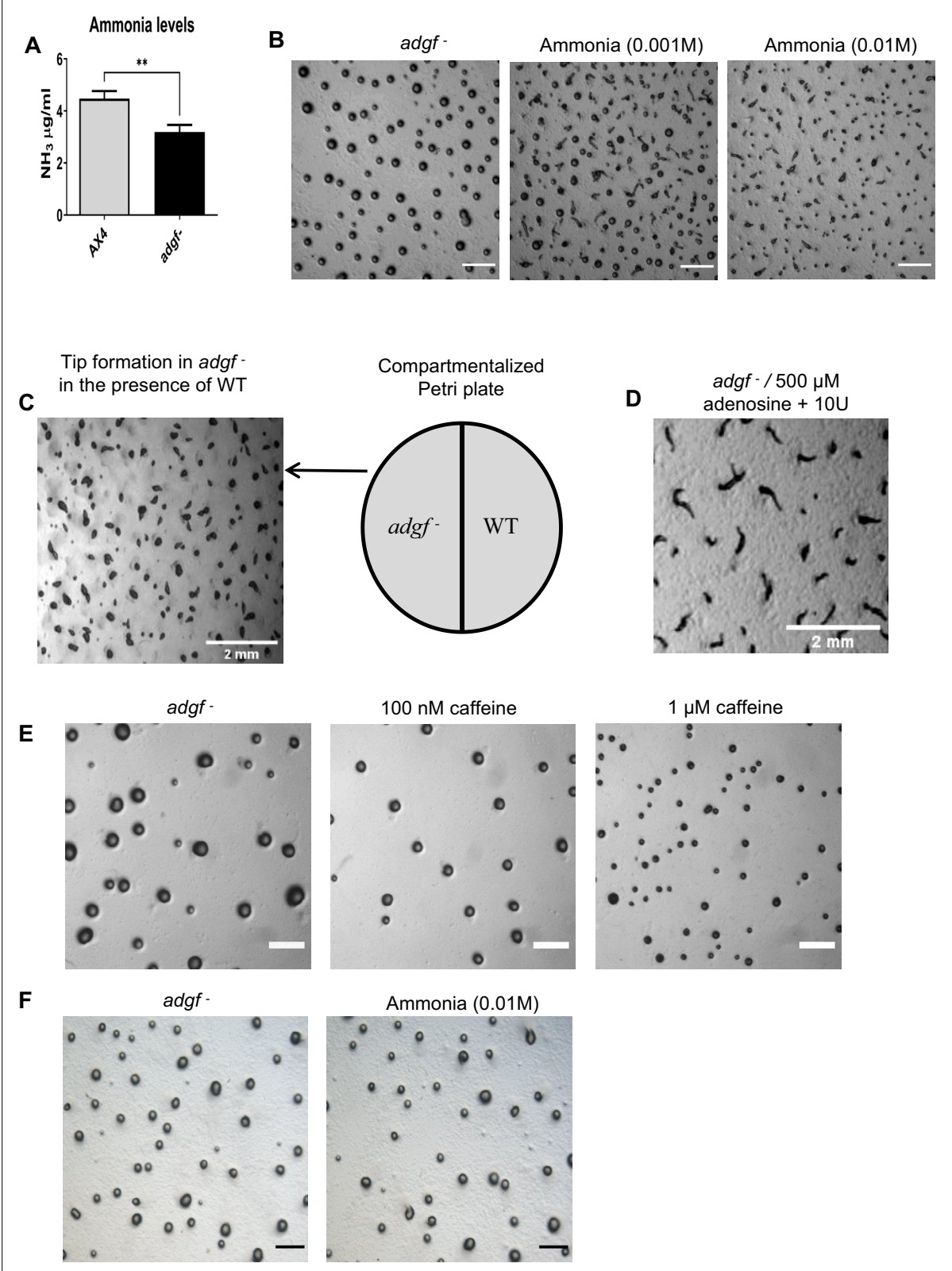

**Figure 5.** Adenosine deamination reaction rescues the mound arrest of *adgf*. (**A**) Quantification of ammonia using the ammonia assay kit. WT and *adgf⁻* mounds were harvested and lysed using a cell lysis buffer. Cell debris was removed by centrifugation, and the supernatant was used to quantify ammonia. Error bars represent mean and SEM (*n*=3). **p<0.01, by Student's *t*-test. (**B**) Treatment of *adgf⁻* mounds with ammonia. Ammonia was generated by mixing 2 ml of NH₄Cl and 2 ml of 1 N NaOH. The mixture was poured on top of the lid and the KK2 plates with the mounds were inverted

*Figure 5 continued on next page*

*Figure 5 continued*

and sealed thereafter. Images were taken 3.5 hr post treatment. Dose-dependent effect of ammonia on the rescue. Scale bar: 2 mm (*n* = 3). (**C**) WT and *adgf*⁻ cells on either side of a compartmentalized Petri dish led to tip formation in *adgf*⁻. Scale bar: 2 mm; (*n*=3). (**D**) *adgf*⁻ cells on one side and KK2 buffer containing adenosine and ADA on the other side of the compartmentalized dish, rescued the mound defect. Caffeine rescues the large mound size of *adgf* mutant. Scale bar: 2 mm; (*n*=3). (**E**) *adgf*⁻ cells were treated with different concentrations of caffeine (100 nM, 1 μM) while plating, and images were taken during mound stage. Scale bar: 2 mm; *n* = 3. (**F**) Exposure to ammonia does not rescue the mound size of *adgf* mutant. *adgf*⁻ mounds were exposed to 0.01 M ammonia and images were captured 3.5 hr post chemical treatment. Scale bar: 2 mm (*n* = 3).

The online version of this article includes the following figure supplement(s) for figure 5:

**Figure supplement 1.** Total protein and RNA levels during mound stage of development.

**Figure supplement 2.** Volatiles released from *K. pneumoniae* rescue the mound arrest phenotype of *adgf*⁻.

**Figure supplement 3.** Treatment with adenosine and other purines does not induce mound arrest in WT.

mutants (*Figure 5F*), suggesting that while some early effects may be mediated through adenosine receptors, the later effects appear to be independent and are likely to be influenced by ammonia.

## High adenosine levels and other related purines do not block tip development

ADA catalyses the conversion of adenosine/deoxy adenosine, respectively, to inosine/deoxy inosine. As *adgf* mutants have high adenosine levels, we investigated if the mound arrest could be mimicked in WT by treating with adenosine. However, the addition of adenosine does not lead to mound arrest in WT (*Figure 5—figure supplement 3A*). It is possible that ADA/ADGF converted adenosine analogues to inosine analogues, which in some manner affected development. Hence, WT cells were treated with adenosine analogue (2'-deoxyadenosine) or guanosine (10 μM each), and they do not cause a mound arrest phenotype either. Similarly, treating *adgf*⁻ mounds with inosine does not restore tip formation (*Figure 5—figure supplement 3B, C*), suggesting that the blocked tip development is due to faulty adenosine deamination alone.

## Faulty expression of cAMP relay genes in *adgf*⁻

cAMP signalling is crucial for tip development and determining cell fate in *Dictyostelium* (*Schaap and Wang, 1986*; *Saxe et al., 1993*; *Firtel, 1996*; *Singer et al., 2019*). Hence, the expression of genes involved in cAMP relay was measured in WT and *adgf*⁻ cells by qRT-PCR. Total cAMP levels and *acaA* gene expression were both low in the *adgf*⁻ lines (*Figure 6A, B*). Treating the cells with the pkA activator 8-Br-cAMP or the activator of adenyl cyclase, cyclic-di-GMP (*Wang and Schaap, 1985*; *Chen and Schaap, 2012*), reversed the *adgf*⁻ mound arrest phenotype (*Figure 6C, D*), and increasing the cyclic-di-GMP dose from 0.5 to 1 mM resulted in multi-tipped phenotype (*Figure 6E*). These observations reinforce that the mound arrest phenotype of the *adgf*⁻ lines is due to faulty cAMP signalling. A transient increase in cAMP levels can be achieved by blocking the phosphodiesterase (*pde4*) activity. When treated with a known PDE4 inhibitor, 3-isobutyl-1-methylxanthine (IBMX) (*Bader et al., 2007*; *Siegert and Weijer, 1989*), there was no effect on tip formation in *adgf*⁻ mounds (*Figure 6F*), although caffeine restored tip formation (*Figure 6G*).

## Circular instead of spiral cAMP waves in *adgf*⁻ mounds

Low *acaA* expression, reduced cAMP levels and enhanced cell–cell contacts in *adgf*⁻ are likely to impair cAMP wave propagation, and to ascertain if this is true, the cAMP signal propagation in WT and *adgf*⁻ mounds was compared using dark field optics. The cAMP wave propagation was spiral in WT, and in contrast, the waves were circular in the *adgf* mutant, suggesting that the cAMP relay is impaired (*Figure 6H*; *Figure 6—videos 1 and 2*). While this strongly suggests a defect in the mutant's ability to sustain spiral wave formation, the possibility of short-lived or transient spiral waves in the mutant cannot be excluded.

## Ammonia restores tip formation by regulating *acaA/pde4* expression

To find if *adgf* expression is regulated by the substrate or the product, WT cells were treated with different concentrations of adenosine (100 nM, 1 μM, and 0.5 mM), plated on KK2 agar plates and harvested at 16 hr for RNA isolation. Independently, WT mounds on KK2 agar plates were exposed

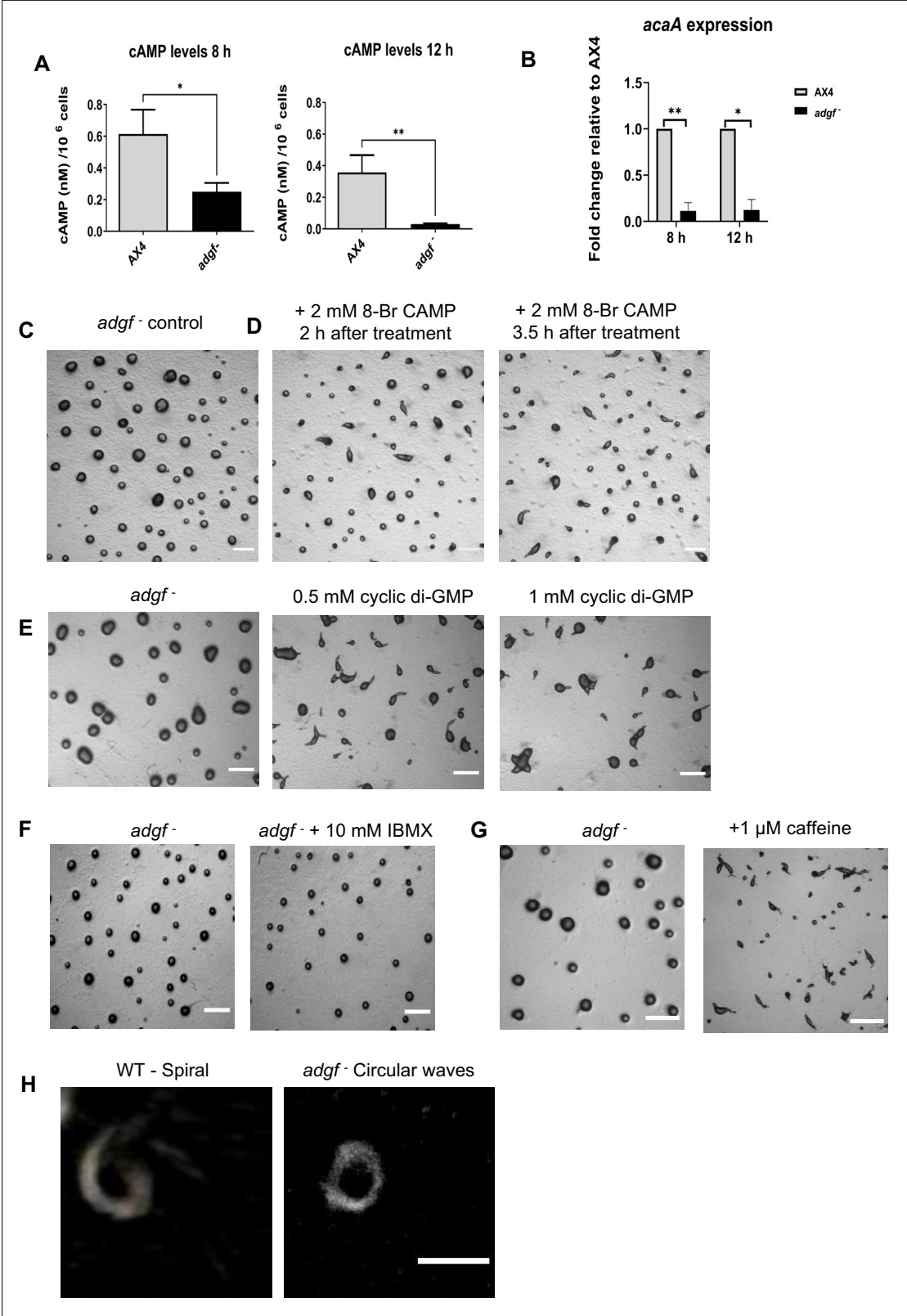

**Figure 6.** Impaired cAMP signalling in *adgf⁻*. (**A**) Total cAMP levels in WT and *adgf⁻* mounds were quantified using cAMP-XP assay kit (Cell Signaling, USA). Level of significance is indicated as *p < 0.05, **p < 0.01, by Student's t-test. Error bars are mean ± SEM (*n* = 3). (**B**) *acaA* expression was quantified using qRT-PCR. The error bars represent the mean ± SEM (*n* = 3). *p< 0.05, **p< 0.01 (Student's t-test). (**C**) *adgf⁻* mounds. (**D**) Time course of *adgf⁻* mounds treated with 8-Br-cAMP and imaged at different intervals. Scale bar: 2 mm (*n* = 3). Treatment with cyclic di-GMP and caffeine rescues

*Figure 6 continued on next page*

*Figure 6 continued*

the mound arrest phenotype. (E) Addition of cyclic-di-GMP restored tip formation in *adgf⁻* 3.5 hr after the treatment. Scale bar: 1 mm (*n* = 3). (F) PDE inhibitor (IBMX) treatment failed to rescue the *adgf⁻* mound arrest. Scale bar 1 mm (*n* = 3). (G) *adgf⁻* mounds treated with caffeine formed tips 3.5 hr post treatment. Scale bar: 2 mm (*n* = 3). Altered cAMP wave pattern in *adgf* mutants. (H) Optical density waves depicting cAMP wave generating centres in WT and *adgf⁻*. WT shows spiral and *adgf⁻* exhibits circular wave propagation. Scale bar 1 mm; (n=3).

The online version of this article includes the following video(s) for figure 6:

**Figure 6—video 1.** Timelapse video of cAMP wave propagation in AX4.

https://elifesciences.org/articles/104855/figures#fig6video1

**Figure 6—video 2.** Timelapse video of cAMP wave propagation in *adgf⁻*.

https://elifesciences.org/articles/104855/figures#fig6video2

to different concentrations of volatile ammonia (0.1, 1, and 10 mM) for 3 hr and thereafter, the mRNA expression levels of *adgf*, *acaA*, and *pde4* were examined. With 100 nM adenosine, the expression of both *adgf* and *acaA* decreased, but at higher adenosine concentrations (1 µM and 0.5 mM), the expression levels of these two genes were comparable to controls (*Figure 7A*). Interestingly, *pde4* expression decreased gradually with increasing adenosine concentrations (100 nM to 0.5 mM), but increased steadily with increasing ammonia levels.

Exposure of WT mounds to 0.1 mM ammonia led to decreased *adgf* expression, but exposure to 1 and 10 mM ammonia, respectively, caused a 3- and 2.3-fold upregulation of *acaA* expression. These observations suggest that ammonia may be rescuing the mound arrest phenotype by enhancing *acaA* expression and thus cAMP levels. In similar conditions, a considerable increase in *pde4* expression was observed (*Figure 7B*), which may be necessary for controlling cAMP levels in response to ammonia treatment. In conclusion, the expression of *adgf* is influenced both by the substrate adenosine and the product ammonia in a dose-dependent manner. To ascertain if ammonia exposure indeed increases the cAMP levels in the mutant, thereby restoring the phenotype, cAMP levels were measured in the *adgf* mutant mounds following the rescue with ammonia. Indeed, the cAMP levels were higher in the rescued mounds compared to untreated *adgf* controls (*Figure 7C*), supporting the idea that ammonia promotes tip development by restoring the cAMP signalling.

## *adgf⁻* cells sort to prestalk in a chimera with WT

To determine if *adgf* favours the differentiation of one cell type over the other, the expression of pst and psp markers was examined in the mutant by realtime PCR. While the expression of pst markers, *ecmA* and *ecmB*, was significantly upregulated, the psp-specific *pspA* gene expression was reduced in the mutant (*Figure 7D*), suggesting that WT *adgf* favours psp expression.

To know the cell type preference of the *adgf* mutants in a chimera with WT, a fraction (20%) of any one cell type was labelled with a cell tracker stain 1,1'-Dioctadecyl-3,3,3',3'-Tetramethylindocarbocyanine Perchlorate (DIL), and the development was tracked thereafter. In the slugs that formed after mixing labelled WT or *adgf*^OE^ cells with 80% unlabelled *adgf⁻* cells, the fluorescence was largely confined to the psp region (*Figure 8B, C*). Conversely, when *adgf⁻* cells stained with DIL were mixed with unlabelled WT or *adgf*^OE^, the fluorescence was restricted to the pst part of the slug (*Figure 8A*). In the chimeric slugs, consistently, labelled WT or *adgf*^OE^ cells occupy the psp, whereas the *adgf⁻* cells end up in the pst region. Thus, the distribution of cell types in the mound/slug is significantly influenced by *adgf*, as the mutant cells sort out in a mixture with WT cells and differentiate to pst cells.

Further, when *adgf⁻* slugs were stained with the pst marker, neutral red (NR) (*Yamamoto and Takeuchi, 1983*), the staining was intense in the pst region and anterior-like cells (ALCs), and such an intense colouration was not apparent in WT especially in the ALCs (*Figure 8—figure supplement 1A*). To further investigate if reduced ADA activity affects cell type specific marker expression or their pattern in slugs, pst (*ecmA*-GFP, *ecmO*-GFP) or psp (*pspA*-RFP) lines were treated with the ADA inhibitor DCF. While there was no change in pst/psp patterns in the fraction of mounds and slugs that developed after a long delay, the *ecmA*-GFP fluorescence was intense when compared to WT (*Figure 8—figure supplement 1B*). Similar to the NR staining pattern in *adgf* mutant slugs, a prominent expression of *ecmA*-GFP was observed in the ALC region also. Visually, there was no change in *ecmO* or *pspA* marker expression in DCF-treated slugs (*Figure 8—figure supplement 1C, D*), suggesting that impaired ADGF activity affects tip-specific expression.

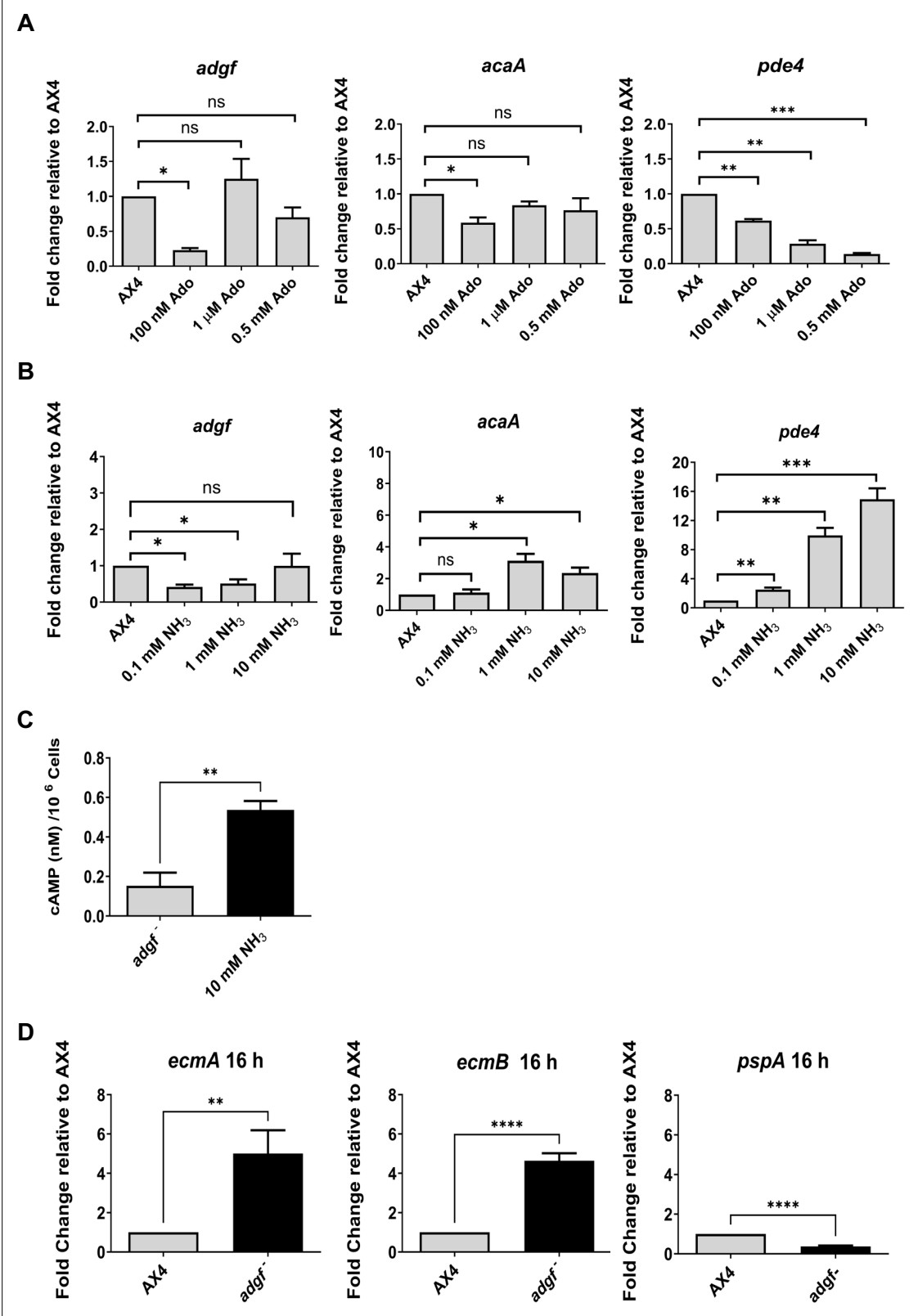

**Figure 7.** Expression levels of *adgf*, *acaA*, and *pde4* in response to adenosine and ammonia treatment. (**A**) Expression levels of *adgf*, *acaA*, and *pde4* in response to adenosine treatment (100 nM, 500 nM, and 1 μM). (**B**) And ammonia treatment (0.1, 1, and 10 mM). Data represent mean and SEM. Level of significance is indicated as *p < 0.05, **p < 0.01, and ***p < 0.001, ns-non significant (*n* = 3) by one way ANOVA analysis. (**C**) cAMP levels in *adgf* mutants rescued with ammonia. Level of significance is indicated as **p < 0.01 (*n* = 3, Student's t-test). Expression levels of prestalk (pst), *ecmA*,

*Figure 7 continued on next page*

*Figure 7 continued*

*ecmB* and prespore (psp), *pspA* cell type markers in *adgf⁻*. The expression profile of (**D**) pst (*ecmA, ecmB*) and psp (*pspA*) specific markers in WT and *adgf⁻* was quantified using qRT-PCR. Error bars represent mean and SEM. Level of significance is indicated as **p < 0.01 and ****p < 0.0001 (*n* = 3) by Student's t-test. The fold-change in RNA transcript levels is relative to WT at the indicated time points. *rnlA* was used as the internal control.

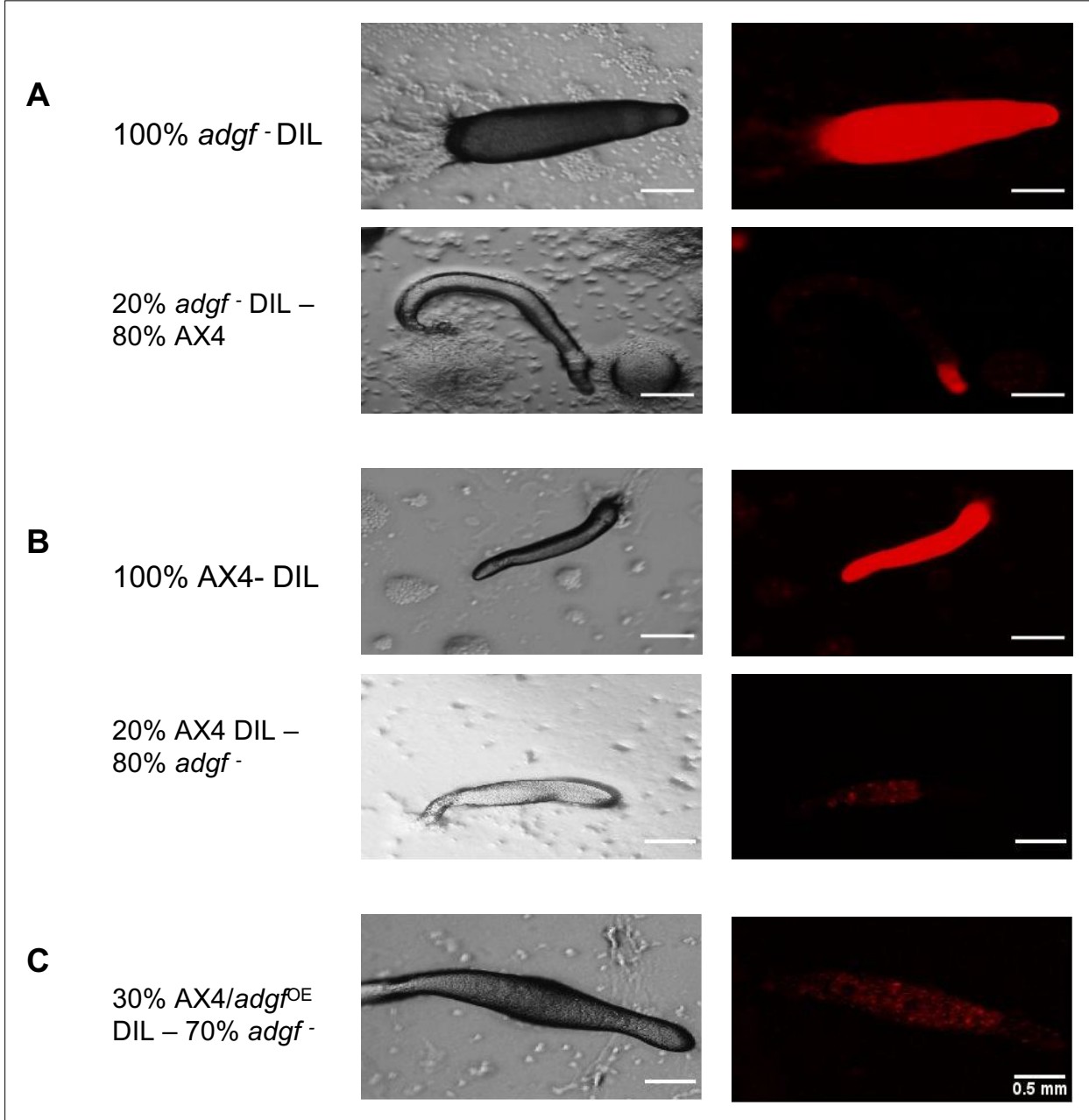

**A** 100% *adgf⁻* DIL

20% *adgf⁻* DIL – 80% AX4

**B** 100% AX4- DIL

20% AX4 DIL – 80% *adgf⁻*

**C** 30% AX4/*adgf*^OE DIL – 70% *adgf⁻*

**Figure 8.** Mixing of WT cells with *adgf⁻* following DIL staining. (**A–C**) DIL labelled cells were mixed with unlabelled cells and plated on KK2 agar. Images were captured during the migrating slug stage. The left panel shows bright field, and the right panel shows the corresponding fluorescence images. Scale bar: 0.5 mm (*n* = 3).

The online version of this article includes the following figure supplement(s) for figure 8:

**Figure supplement 1.** Neutral red staining of mounds and slugs.

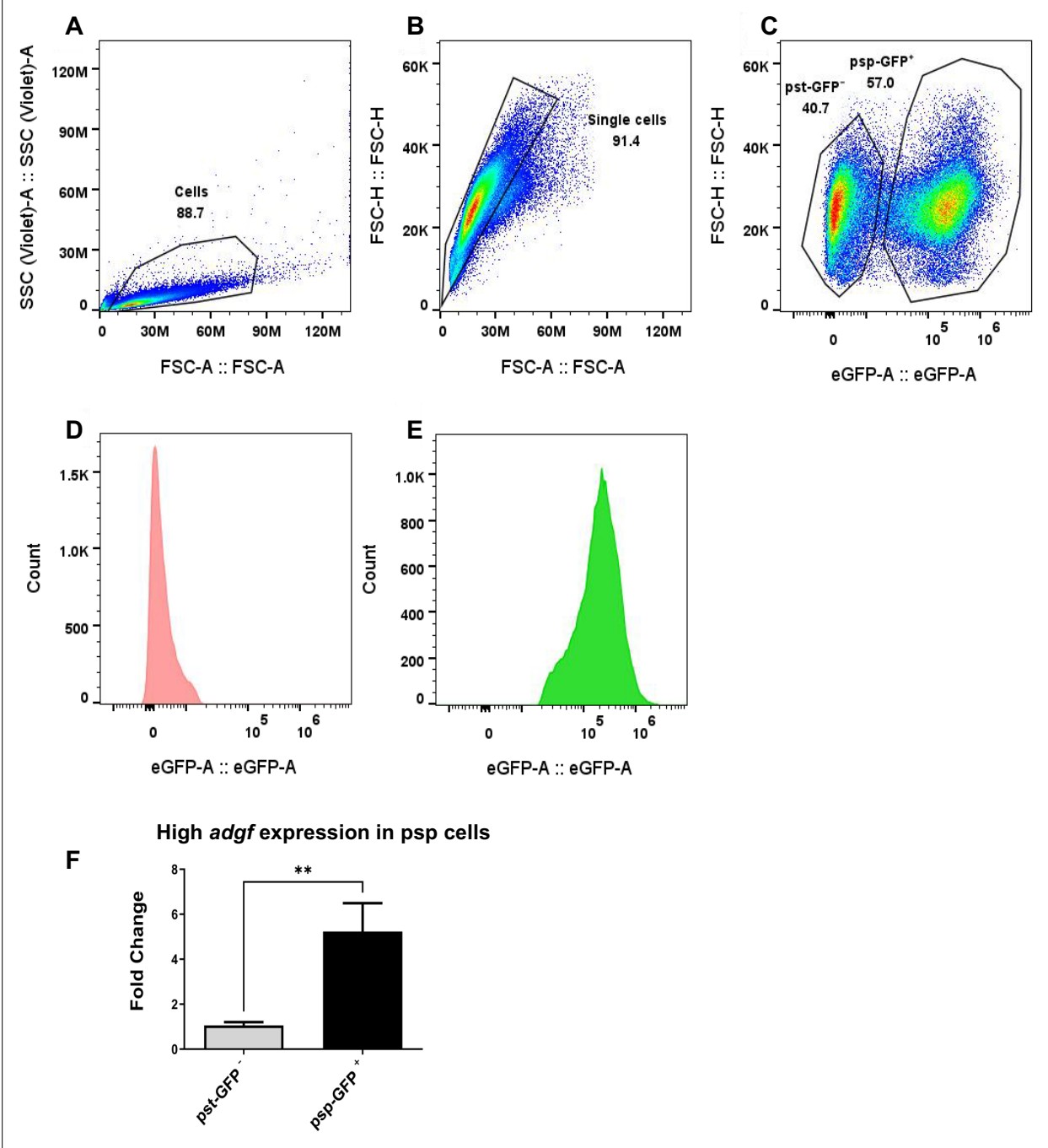

**Figure 9.** Sorting of pst-GFP- and psp-GFP+ *Dictyostelium* cells by fluorescence activated cell sorter. (**A**) Forward scatter (FSC) vs. side scatter (SSC) plot used to gate total cells. (**B**) Singlets were gated based on FSC-H vs. FSC-A to exclude doublets and aggregates. (**C**) GFP fluorescence profile of gated single cells reveals two populations: pst-GFP⁻ and psp-GFP⁺, corresponding to pst and psp cells, respectively. (**D**) GFP⁻ (pst) and (**E**) GFP⁺ (psp) fractions. (**F**) *adgf* expression in FACS sorted samples was quantified by quantitative real-time PCR (qRT-PCR). Error bars are mean ± SEM. Level of significance is indicated as **p < 0.01 (*n* = 3) by Student's t-test.

To know if *adgf* expression is differentially regulated between the two major cell types, psp-GFP tagged WT slugs were disaggregated, and using the fluorescence activated cell sorter (FACS), GFP-negative prestalk (pst-GFP⁻) and GFP-positive prespore (psp-GFP⁺) populations were obtained (*Figure 9A–E*), and RNA was isolated from both cell populations. Real-time PCR analysis reveals that *adgf* expression is 4.95-fold higher in the psp-GFP⁺ than pst-GFP⁻ cells (*Figure 9F*). The preferential sorting of WT cells to the psp region and the *adgf* mutants (with reduced psp expression) to the

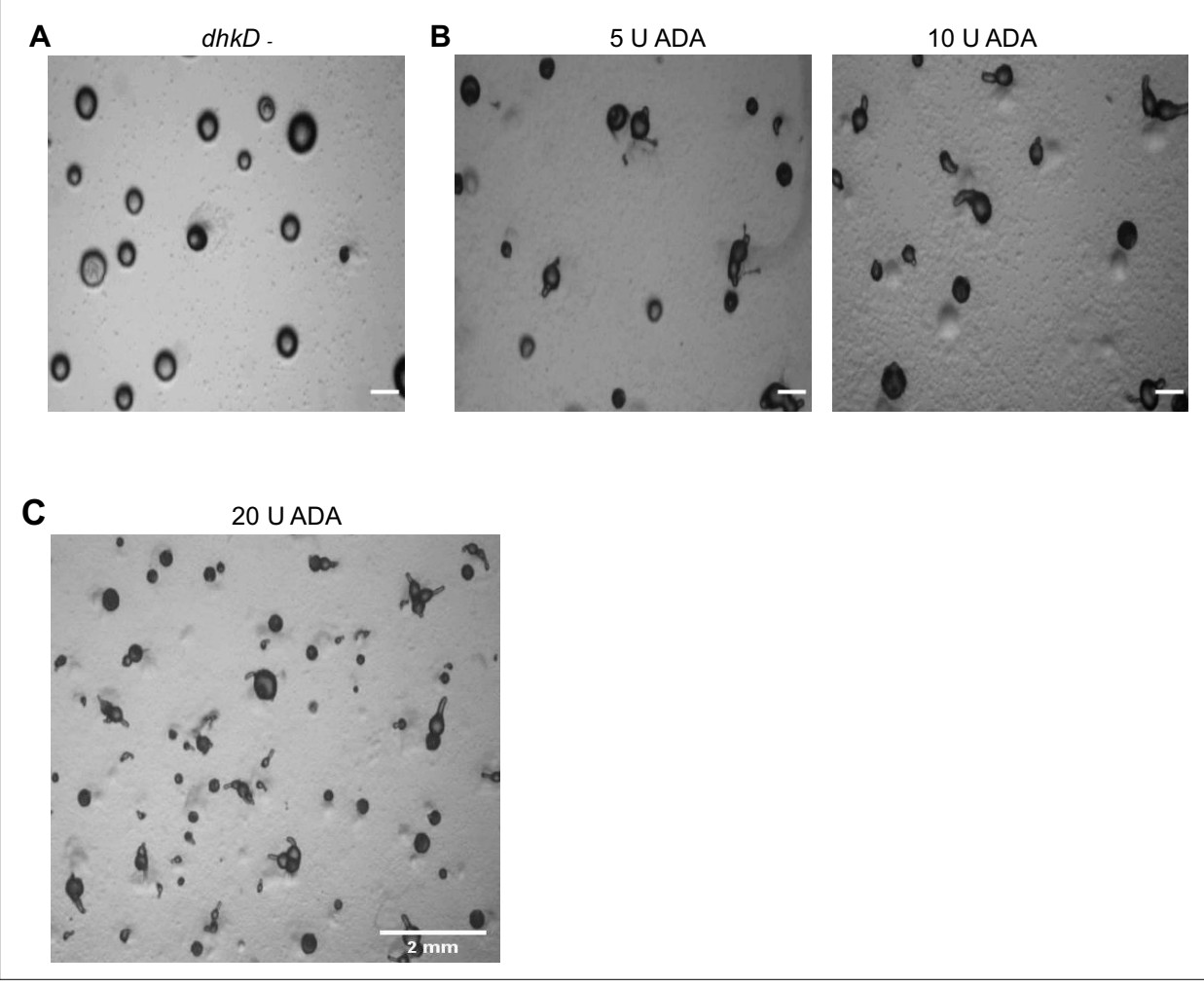

**Figure 10.** *adgf* acts downstream of the histidine kinase *dhkD*. (**A**) *dhkD* mutants on KK2 agar plates. Scale bar 1 mm; (*n*=3). (**B**) 5 and 10 U ADA rescued the mound arrest phenotype in a dose-dependent manner. Scale bar: 1 mm (*n* = 3). Images were taken 3.5 hr post treatment. (**C**) Addition of 20 U ADA led to formation of multiple tips. Scale bar: 2 mm (*n* = 3).

The online version of this article includes the following figure supplement(s) for figure 10:

**Figure supplement 1.** Treatment of mounds with ADA and DCF.

**Figure supplement 2.** Developmental phenotype of different deaminase gene knockouts.

pst part of the slug also reinforces the differential expression of *adgf*. Thus, cells with higher *adgf* expression preferentially sorted to the psp region, and the absence of *adgf* in mutant cells may hinder their ability to adopt a prespore fate, leading to their preferential sorting in the pst region within the chimera.

### *adgf* acts downstream of the histidine kinase, *dhkD*

In an effort to identify the pathway by which *adgf* acts during tip development, we selected a number of mutant lines with similar mound arrest phenotype such as cAMP receptor B (*carB*⁻), LIM domain containing protein (*limB*⁻), mound mutant (*mndA*⁻) and histidine kinase (*dhkD*⁻) (*Saxe et al., 1993*; *Carrin et al., 1996*; *Chien et al., 2000*; *Singleton and Xiong, 2013*), and treated the mounds with ADA enzyme. Treatment with 10 U ADA or exposure to ammonia restored tip formation in *dhkD*⁻ mounds and not others tested (*Figure 10—figure supplement 1A, B*). An increase in ADA concentration (20 U) resulted in the development of multiple tips in *dhkD*⁻ (*Figure 10A–C*). Further, in the *dhkD* mutant mounds that were rescued by ammonia, cAMP levels were significantly elevated compared to untreated *dhkD*⁻ controls (*Figure 10—figure supplement 1C*). These findings imply

that ammonia-induced tip formation increased cAMP levels, thereby restoring the phenotype, and *adgf* acts downstream of *dhkD* in controlling tip development. Similarly, we tested a few mutants with multi-tipped phenotype, such as tipped mutant (*tipA*⁻), culinB (*culB*⁻), autophagy mutants (*atg7*⁻, *atg8*⁻, and *atg9*⁻) (*Stege et al., 1999*; *Wang and Kuspa, 2002*; *Otto et al., 2003*; *Otto et al., 2004*; *Tung et al., 2010*) by adding the ADA inhibitor, DCF (1 mM) to the cell suspension/agar plates and if rescued, those mutants are likely to be in the same pathway as that of *adgf* in controlling tip development. However, DCF treatment had no impact on the phenotype of the mutants with multiple tips (*Figure 10—figure supplement 1D*).

## Impaired expression of other deaminases also results in aberrant tip formation

Several pathways control ammonia levels during development, and knockouts in other deaminases (2-aminomuconate deaminase: DDB_G0275081, adenosine monophosphate deaminase: DDB_G0292266, dCTP deaminase: DDB_G0293580, threonine deaminase: DDB_G0277245, *N*-acetyl glucosamine deaminase: DDB_G0286195, glucosamine-6-phosphate deaminase: DDB_G0278873) show a partial mound arrest phenotype (*Figure 10—figure supplement 2*) although the expression of some of these candidates is stronger during development, suggesting a prominent and unique role of adenosine deamination in tip development.

## Discussion

### *Dictyostelium* ADGF is likely to be a secreted growth factor

Multiple sequence alignments, experiments with CM and cell mixing with WT suggest that ADGF is secreted. ADGF is also known to interact with 5'-adenosine monophosphate (AMP) and deoxyadenosine. AMP deaminase, previously characterized in the mollusk *Helix pomatia*, has been identified as a member of ADGF family (*Tzertzinis et al., 2023*). Studies on the characterization and expression of ADGF in Pacific abalone have shown that it is a secreted protein critical for embryonic and larval development (*Hanif et al., 2022*). The crystal structure of human ADA2 reveals the presence of a catalytic- and two unique ADA2-specific domains with novel folds, responsible for protein dimerization and interaction with cell surface receptors. Furthermore, the presence of a number of *N*-glycosylation sites, conserved disulphide bond, and a signal peptide indicate that ADA2 is specifically adapted to function in the extracellular environment (*Zavialov et al., 2010*).

### Possible reasons for increased mound size in the *adgf* mutant

Enhanced cell adhesion influences cohesion and can impact the mound size (*Roisin-Bouffay et al., 2000*). The expression of *cadA* and *csaA* genes was upregulated in the *adgf* mutant, possibly leading to increased adhesion and larger mound formation. While over-secretion of *ctn* leads to stream breaking and small aggregate formation (*Brock and Gomer, 1999*), disruption of the *ctn* gene prevents this process, inducing large aggregate formation. *ctn* in turn regulates *smlA* (*Brock et al., 2003*), impacting the overall aggregate size. Thus, *adgf* mutants that have reduced *ctn* and *smlA* expression manifest in large mound formation. Indeed, cells treated with adenosine are known to form large aggregates (*Schaap and Wang, 1986*, *Jaiswal et al., 2012*), and the first genetic evidence supporting the previous work is from *adgf* mutants, which carry excess extracellular adenosine and form large aggregation streams.

### Pathways generating ammonia in *Dictyostelium*

Ammonia can come from a variety of sources both within and outside the cells and this can be from dead cells also. Proteolysis during starvation is believed to be the main source of volatile ammonia in *Dictyostelium* (*Hames and Ashworth, 1974*; *Schindler and Sussman, 1977*; *White and Sussman, 1961*), while RNA degradation is also attributed to yield ammonia during starvation (*Walsh and Wright, 1978*). During development, amino acids, total protein and RNA levels are reported to reduce with a significant increase in ammonia levels, thus equivocating the source of volatile ammonia (*Hames and Ashworth, 1974*). The highly acidic, autophagic vesicles in pst cells (*Gross, 2009*) are believed to catalyse the breakdown of proteins and RNA, also generating ammonia.

Furthermore, several deaminases or enzymatic reactions in *Dictyostelium* may also generate ammonia (*Supplementary file 1A*). The annotated *D. discoideum* genome contains five evolutionarily conserved families of ammonium transporter/methylammonium permease/rhesus protein (Amt/Mep/Rh) encoding genes (*Eichinger et al., 2005*), *amtA*, *amtB*, *amtC*, *rhgA*, and *rhgB*, which help in regulating ammonia levels during growth and development. *amtA* and *amtC* antagonistically control developmental processes and are involved in ammonium sensing or transport (*Follstaedt et al., 2003*; *Kirsten et al., 2005*; *Singleton et al., 2006*).

## Role of ammonia during *Dictyostelium* development

Ammonia is known to inhibit aggregation (*Schindler and Sussman, 1979*; *Williams et al., 1984*), affect aggregate territory size (*Thadani et al., 1977*), aggregate density (*Feit, 1988*), cell fate (*Gross et al., 1988*), culmination (*Davies et al., 1993*), and fruiting body size in *Dictyostelium* (*Lonski, 1976*). Ammonia promotes psp over pst differentiation (*Newell et al., 1969*; *Sternfeld and David, 1979*; *Gross et al., 1983*; *Oyama and Blumberg, 1986*), and favours prolonged slug migration called 'slugging' over culmination by suppressing DIF biosynthesis (*Neave et al., 1983*). Ammonia plays a crucial role in orienting cell masses, accelerating the movement of aggregating cells (*Bonner et al., 1986*), and preventing the developmental transition of slugs to fruiting bodies (*Bradbury and Gross, 1989*; *Wang and Schaap, 1989*). Enzymatic removal of ammonia causes the quick transition from slug to fruiting, thus controlling the morphogenetic pathways (*Schindler and Sussman, 1977*).

Beyond its broad effects on aggregation and culmination, ammonia also reinforces positional information by elevating intracellular cAMP levels, favouring prespore over prestalk differentiation (*Bradbury and Gross, 1989*; *Riley and Barclay, 1990*; *Hopper et al., 1993*). Ammonia is known to influence rapid patterning of *Dictyostelium* cells confined in a restricted environment (*Sawai et al., 2002*). In *adgf* mutants that have low ammonia levels, both NR staining and the prestalk marker *ecmA/ecmB* expression are higher than the WT, and the mound arrest phenotype can be reversed by exposing the *adgf* mutant mounds to ammonia. As a gas, ammonia can diffuse generating a gradient. The slime sheath at the back of the slug is believed to prevent the diffusion of ammonia (*Gross, 2009*), and ammonia escaping through the slug front neutralizes the acidic vesicles in prestalk cells (*Bonner, 1952*). Thus, a rise in the pH of these acidic vesicles and the cytoplasm (*Poole and Ohkuma, 1981*; *Gross et al., 1983*; *Davies et al., 1993*) is known to increase the speed of chemotaxing amoebae (*Siegert and Weijer, 1989*; *Van Duijn and Inouye, 1991*), favouring collective cell movement (*Bonner et al., 1988*; *Bonner et al., 1989*), and tipped mound development. Thus, ammonia actively promotes the transition from mound to tipped mound formation.

Our results (*Figure 6A*) also show that the amount of ammonia released from adenosine is in the same order of magnitude as that from other sources (*Yoshino et al., 2007*). It is interesting that a mutation in *adgf* manifests in arrested tip development, although *adgf* expression was found to be higher in psp than pst cells. If a threshold concentration of ammonia is not present, collective cell movement influencing tip formation may be blocked at the mound stage. Increased *adgf* expression in psp than pst cell types suggests that low extracellular adenosine (*Weijer and Durston, 1985*; *Schaap and Wang, 1986*) and high levels of ammonia influence psp cell fate. The decision between the formation of the tip (pst cells) and the ALC is controlled by the tip's production of ammonia, which prevents the migration of ALCs towards the tip (*Sternfeld and David, 1982*; *Feit et al., 1990*).

In slugger mutants, ammonia is known to inhibit tip formation (*Gee et al., 1994*), but not in WT NC4. *Yoshino et al., 2007* have also reported ammonia inhibiting both aggregation and tip formation in WT cells. However, ammonium chloride was used as a source of ammonia, and the potential interference from chloride ions cannot be ruled out. A mutant impaired in mitochondrial function, *midA⁻*, is reported to accumulate high levels of ammonia that inhibited development (*Torija et al., 2006*). Notably, these cells were developed with overhead light, a condition that may influence ammonia sensitivity and the developmental phenotype. However, ammonia exerts no effect on tip formation in AX4 (data not shown). After the tips are established, the slugs/fruiting bodies move away from ammonia, reflecting a process by which fruiting bodies position themselves from other structures to increase the possibility of spore dispersal (*Kosugi and Inouye, 1989*). A gaseous signal can act over long distances in a short time and, for instance, ammonia promotes synchronous development in a colony of yeast cells (*Palková et al., 1997*; *Palková and Forstová, 2000*). The slug tip is known to release ammonia, probably favouring synchronized development of the entire colony of *Dictyostelium*.

However, after the tips are established, ammonia exerts negative chemotaxis, probably helping the slugs to move away from each other, ensuring equal spacing of the fruiting bodies (**Feit and Sollitto, 1987**). Taken together, these findings suggest that ammonia acts as both a local and long-range regulatory signal, integrating environmental and cellular cues to coordinate multicellular development.

Cells carrying high ATP, its derivatives cAMP (**Bagorda et al., 2009**; **Singer et al., 2019**) and adenosine (**Schaap and Wang, 1986**), end up in the tip (**Hiraoka et al., 2022**). Further, 5′-nucleotidase promoter activity is high in pstAB cells (**Ubeidat et al., 2002**) that significantly covers the tip region, suggesting high adenosine levels in the tip. However, extracellular 5′-AMP can be derived from multiple sources such as hydrolysis of cAMP, ATP, and RNA (**Carpousis et al., 1999**).

In *adgf* mutants, ammonia levels may not be sufficient enough to neutralize the acidic vesicles and hence, NR staining is intense in the pst region and the ALCs (**Bonner, 1952**). Ammonia has been shown to differentially suppress cAMP chemotaxis in ALC and pst cells in *Dictyostelium* (**Feit et al., 2001**).

## Ammonia's effect on cAMP signalling in *Dictyostelium*

Exposing the cells to ammonia is known to increase intracellular cAMP levels in *D. discoideum* (**Riley and Barclay, 1990**; **Feit et al., 2001**), but **Schindler and Sussman, 1977** and **Williams et al., 1984** reported that high ammonia levels inhibit the synthesis and release of cAMP. However, these experiments use ammonium carbonate (**Schindler and Sussman, 1977**), or ammonium chloride (**Williams et al., 1984**) as a source of ammonia, and the possibility of carbonate and chloride ions interfering with development cannot be ruled out. Ammonia is reported to show no effect on cAMP levels in *D. discoideum* slugs (**Schaap et al., 1995**), but in *D. mucoroides,* high ammonia levels are known to block the production of extracellular cAMP. The effect of ammonia on aggregation seems to be species specific, notably in *P. violaceum*, *P. pallidum*, *and D. mucoroides,* resulting in wider aggregation territories, while it was not observed in other species such as *D. discoideum* and *D. purpureum* (**Bonner and Hoffman, 1963**).

Adenylate cyclase (AC) activity is regulated by various factors (**Steer, 1975**). At physiological levels, ammonium ions increase AC activity in the rat brain by 40% (**Yeung et al., 1989**), but lowered its activity in the liver by about 30% (**Wiechetek et al., 1979**). Ammonia has been shown to increase the activity of AC-G in *Dictyostelium* sori (**Cotter et al., 1999**). High ammonia levels by altering the pH can affect the activity of numerous enzymes whose activity is pH dependent and thus the activity of AC can be impacted by pH variations. *adgf* mutants with low ammonia levels have reduced cAMP levels, and an increase in ammonia causes a significant increase in *aca*A expression. In rat brain, ammonia is known to interact with manganese (**Rivera-Mancía et al., 2012**; **Lu et al., 2020**), which is known to increase *aca*A expression in *Dictyostelium* (**Loomis et al., 1979**; **Khachatrian et al., 1987**). Possibly, ammonia interacts with metal ions like manganese to form 'ammine complexes' (**Lipkowski and Galus, 1973**), particularly in aqueous environments or under specific conditions where the appropriate ligands are available, and enhances cAMP levels, thus rescuing the mound defects of the mutant.

Tip organizer development in *Dictyostelium* depends on the differentiation and appropriate sorting of pst and psp cells (**Kay et al., 1978**; **Williams et al., 1989**; **Saxe et al., 1993**; **Williams, 2006**), a process that relies on signalling of different morphogens including cAMP, adenosine, DIF and ammonia (**Bloom and Kay, 1988**; **Williams, 1988**; **Riley and Barclay, 1990**; **Gross, 1994**). These morphogens modify one another's effects and determine the choice of the differentiation pathway as well as the spatial arrangement of cells (**Bloom and Kay, 1988**). Thus, the rescue of the *adgf* mutant upon exposure to ammonia is likely due to cAMP signalling and cell–cell contact.

## Possible reasons for reduced cAMP levels in the *adgf* mutant

Higher *pde4* and reduced *acaA* expression may result in low cAMP levels in the mutant. ADA significantly regulates adenosine levels, thus reducing the activation of adenosine-mediated receptors (**Van Haastert, 1983**). Two classes of adenosine receptors, including adenosine alpha- and beta-receptors, are known to be expressed in *Dictyostelium* (**Theibert and Devreotes, 1984**). When adenosine concentrations are high, beta-receptors bound with adenosine inhibit the binding of cAMP to its receptors, thereby inhibiting cAMP signalling (**Newell, 1982**; **Van Haastert, 1983**; **Theibert and Devreotes, 1984**). High adenosine levels in the *adgf* mutant may also reduce cAMP levels via a similar mechanism.

The mound/slug tip is believed to carry high adenosine levels restricting additional tip formation (*Wang and Schaap, 1985*), and our results also suggest that *adgf*⁻ cells with high adenosine end up as pst cells.

## Distorted cAMP waves in mutant

Collective cell movement within mounds is essential for cell sorting and the progression from mounds to finger structures (*Kellerman and McNally, 1999*). In WT mounds, cAMP waves propagate as spirals, while in the mutants, the wave propagation was in concentric circles. Initially, the cAMP waves arise as concentric circles that, upon symmetry breaking, form spiral waves (*Siegert and Weijer, 1995*). With successive wave propagation, the circular ring distorts, and one end curls towards the pulsatile centre, creating a spiral wave around the organizing centre. In *adgf* mutants, however, defective cAMP relay prevents this transition. A surge in phosphodiesterase inhibition is thought to regulate this shift from concentric to spiral wave propagation (*Pálsson and Cox, 1996*), and indeed the phosphodiesterase expression was higher in the *adgf* mutant, possibly distorting the wave propagation.

The mound/slug tip of *Dictyostelium* generates cAMP pulses (*Traynor et al., 1992*) and is known to suppress additional tip formation (*Farnsworth, 1973*). Adenosine, which inhibits additional tip formation, represses *pde4* expression, whereas ammonia, which promotes tip elongation, increases *pde4* expression. Thus, the restoration of cAMP signalling and spiral wave propagation possibly leads to the rescue of the *adgf*⁻ mound arrest phenotype upon exposure to ammonia. Surprisingly, adenosine exerts little effect on tip development in WT cells, and studies of *Inouye, 1989* also reinforce our observations, showing that adenosine does not significantly influence the conversion of pst to psp cells in shaking culture conditions.

A robust cAMP relay is required for the transition from concentric to spiral wave propagation, with the corresponding changes in AC or phosphodiesterase activity (*Siegert and Weijer, 1995*; *Pálsson and Cox, 1996*). Reduced *acaA* expression in the mutant could bias the wave propagation towards concentric rather than spiral geometry. However, further experiments will be required to ascertain this link.

## Adenosine deamination drives tip organizer development

A crucial evidence supporting that adenosine deamination is singularly responsible for the rescue of *adgf*⁻ mound arrest comes from experiments using a partitioned dish, with a buffered solution containing adenosine on one side, and the mutants on the other half of the dish. Addition of ADA to the buffer containing adenosine resulted in *adgf* mutants forming tipped mounds. Although ammonia's role in *Dictyostelium* development is well established, this report is the first to show a novel role of ammonia in tip development. If natural variants of *Dictyostelium* that fail to form tips exist in soil ecosystems, it is conceivable that these strains could still form fruiting bodies when in proximity to WT strains. This interaction would likely occur through volatile signals such as ammonia diffusing through the shared environment. These diffusible signals may help rescue or coordinate developmental processes among neighbouring cells. Notably, the development of an organism in its natural environment is strongly influenced by volatile compounds present in its surroundings (*Schulz-Bohm et al., 2017*).

## ADA in organizer development and gastrulation in vertebrates

*ada/adgf* expression is found to be high during gastrulation stages in several vertebrates (*Pijuan-Sala et al., 2019*; *Tyser et al., 2021*), and collective cell migration during gastrulation and collective cell movement within the *Dictyostelium* mound are remarkably similar (*Weijer, 2009*), suggesting an overlooked role of ammonia in organizer development. The human homologue of *adgf*, CECR1, is a potential gene for the genetic disorder, Cat-eye syndrome and thus, *Dictyostelium* may also serve as a model to study this condition. Ammonia is crucial in regulating differentiation, metabolism, and gene expression (*Stein et al., 2013*; *Liu et al., 2022*). Ammonium has been shown to induce aberrant blastocyst differentiation, disrupt normal metabolic processes, alter pH regulation, and subsequently affect fetal development in mice (*Lane and Gardner, 2003*). Newt *Triturus* exposed to ammonia shows dorsalization of the ventral marginal zone highlighting the capacity of ammonia to alter embryonic axis formation (*Yamada, 1950*). Furthermore, research on avian embryos has revealed significant ammonia content in developing eggs, suggesting a potential role for ammonia in the energy metabolism during

ontogenesis (*Needham, 1926*). While this raises the possibility of conserved mechanisms, further studies will be needed to determine whether adenosine deamination plays a role in organizer development across different systems.

### *dhkD* functions upstream of *adgf*

The histidine kinase *dhkD*, a member of the two-component family of histidine kinases, appears to function upstream of *adgf*, adding to our understanding of the signalling cascade in *Dictyostelium*. Histidine kinases play essential roles in regulating developmental transitions, with *dhkC*, for example, initiating the late developmental program that ultimately leads to fruiting body formation (*Singleton et al., 1998*). This function of *dhkC* is regulated in part by ammonia levels, as low ammonia concentrations have been shown to inhibit the *dhkC* phospho-relay, thereby influencing the developmental outcomes (*Kirsten et al., 2005*). In *Dictyostelium*, several histidine kinases, including *dhkA*, *dhkB*, *dhkC*, and *dokA*, coordinate cAMP signalling to modulate the activity of PKA, a cAMP-dependent protein kinase that is essential for proper development (*Anjard and Loomis, 2003*). These findings suggest the interdependencies within the signalling network in regulating multicellular development in *Dictyostelium*.

### Model of ADGF action

Ammonia generated from multiple sources is likely to be quenched rapidly by the intracellular acidic vesicles of pst cells, increasing its pH and favouring collective cell movement driving mound/slug tip development. However, the slime sheath at the back of the slug may prevent the diffusion of ammonia, and thus the major route of ammonia emission will be from the slug front (*Farnsworth and Loomis, 1974*). Although ammonia is likely to be formed from several sources in *Dictyostelium*, a critical threshold concentration of ammonia may be necessary for tip development. The failure of even one pathway may result in a drop-in ammonia levels, exerting an effect on development (*Figure 11*). Thus, ammonia is likely to be important for the maturation of tip cells.

Our study shows that *adgf* acts downstream of *dhkD* and in silico studies have shown that *dhkD* interacts with *adgf*. Therefore, it is likely that in the presence of ammonia, *dhkD* activates the phosphorelay, in turn increasing intracellular cAMP levels, leading to tip formation. Histidine kinase *dhkC* is known to phosphorylate *regA* in *Dictyostelium* (*Thomason et al., 1999*). However, based on docking, we found no direct interaction of *regA* with ADGF (data not shown). Thus, the integration of histidine kinase signalling ensures coordinated multicellular organization in *Dictyostelium* (*Aoki et al., 2020*). Previous work in other systems has found ADA to be interacting with CD 26 (dpp IV) (*Tanaka et al., 1993*), and docking DdADGF with AprA (dpp8) also supports this observation. However, adding ADA inhibitor to *aprA* mutant (dpp8) showed no effect on the phenotype (data not shown).

### Supplementary results

#### Genes involved in autophagy and tipped mound formation do not show altered expression in the *adgf* mutant

Autophagy helps in maintaining cellular homeostasis by promoting the breakdown of damaged proteins and organelles (*Mizushima et al., 2008*), and ammonia is recognized to be a diffusible regulator of autophagy in human cells (*Eng et al., 2010*). To investigate if autophagy is impaired in the mutant, the expression of autophagy markers *atg8* and *atg18* was examined and there was no discernible change in the expression levels when compared to WT. The tipless to tipped mound transition, as well as late developmental gene expression, is regulated by *gbf*A, a G-box binding factor (GBF). However, in the *adgf* mutant, the *gbf*A expression levels were comparable with no significant difference between the WT at 16 hr (data not shown). Without the assay for autophagy as well as the assay for GBF activity, these expression studies alone are not conclusive to rule out the role of these factors in tip formation.

#### ADA does not deaminate cytokinin

ADA is known to interact with 5'-AMP (*Hanif et al., 2022*) and in *Dictyostelium*, 5'-AMP serves as a direct precursor of cytokinin, playing a significant role in cellular signalling and development (*Taya et al., 1978*). Although docking studies suggest low to moderate binding of ADGF with cytokinins (zeatin, dihydrozeatin, isopentenyl adenine), we carried out a functional assay by mixing cytokinin and

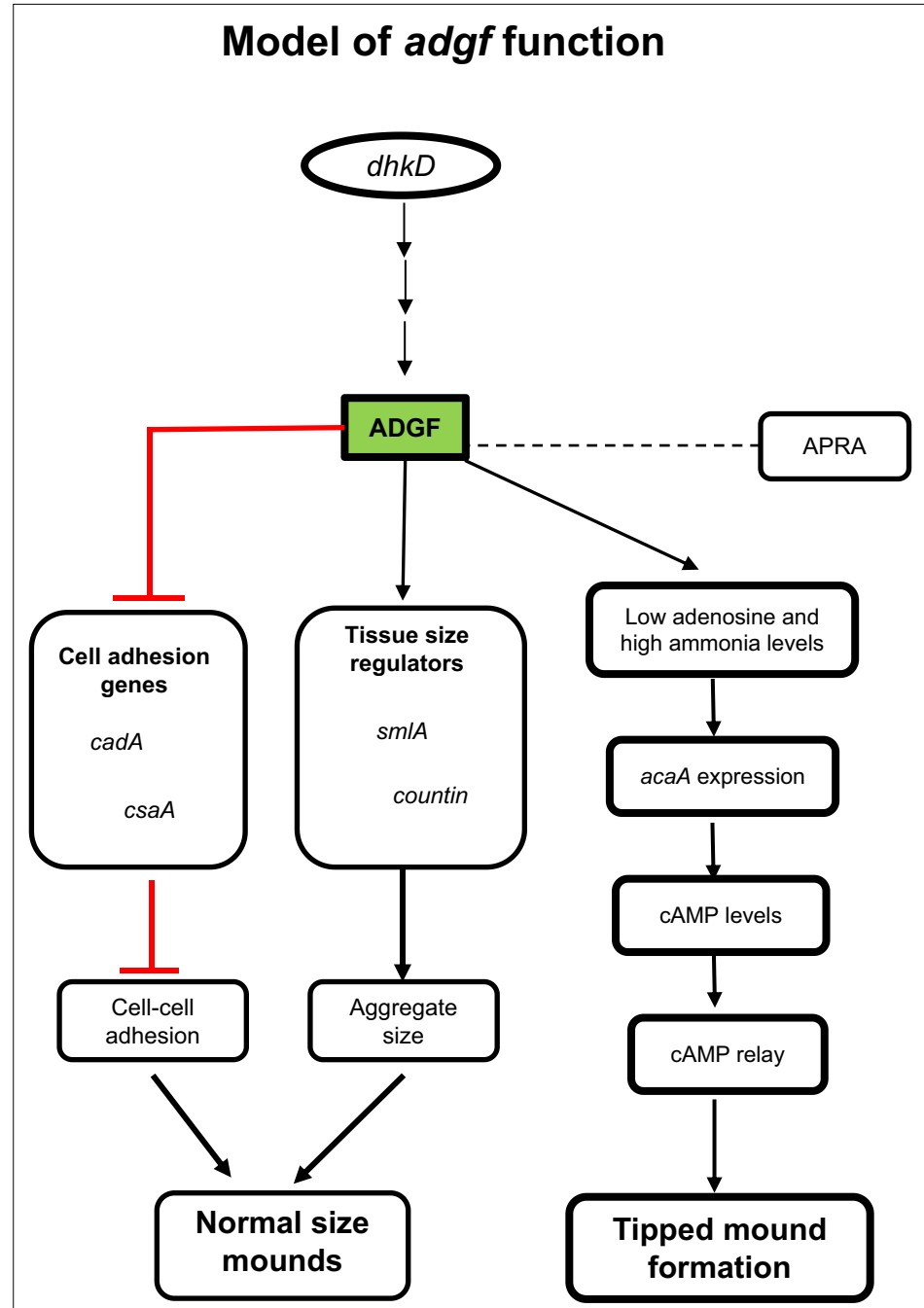

**Figure 11.** Model illustrating the role of *adgf* in development. *adgf* suppresses the expression of genes involved in cell adhesion, *cadA* and *csaA,* and regulates the mound size and tip development by directly acting on adenosine, ammonia levels and cAMP signalling. Line ending in an arrow indicates that the previous gene/factor either directly or indirectly raises the activity or levels of the second; line ending in a cross-bar indicates inhibition. Dotted lines indicate ADGF interacting with APRA.

ADA with KK2 buffer in one side of the Petri dish and found no rescue of the mutant in such conditions (data not shown), suggesting no functional activity between ADA and the versions of cytokinin tested.

## Supplementary discussion

### Plausible routes of caffeine action rescuing the mound arrest phenotype

The *adgf* mound arrest could be rescued by the addition of the adenosine antagonist, caffeine, but treatment with the *pde4* inhibitor, IBMX, failed to rescue the mound arrest. Caffeine is known to reduce adenosine levels in blood plasma (*Conlay et al., 1988*) and also increase ammonium levels in urine samples of rabbits (*Bernheim and Bernheim, 1945*). Thus, the mound rescue upon caffeine treatment may be a result of reduced adenosine and increased ammonia levels. With respect to caffeine action on cAMP levels, the reports are contradictory. Caffeine has been reported to increase AC expression, thereby increasing cAMP levels (*Hagmann, 1986*), whereas *Alvarez-Curto et al., 2007* found that caffeine reduced intracellular cAMP levels in *Dictyostelium*. Although caffeine is moderately potent in inhibiting PDE enzyme activity, the in vivo concentrations are likely to be low to be associated with effective PDE inhibition (*Burg and Werner, 1975*; *Daly, 1993*).

# Materials and methods

## Key resources table

| Reagent type (species) or resource | Designation | Source or reference | Identifiers | Additional information |
|---|---|---|---|---|
| Gene (*Dictyostelium discoideum*) | *adgf* - (adenosine deaminase-related growth factor) | GWDI Bank, Dicty Stock Center, Northwestern University | DBS0237637 (GWDI_47_C_1) | Insertion in exon 2 of DDB_G0275179 |
| Strain, strain background (*Dictyostelium discoideum*) | AX4 (wild-type) | Dictybase (http://dictybase.org/) | DBS0235521 | Parent strain used for all experiments |
| Strain, strain background | *Klebsiella pneumoniae* | Dictybase | DBS0351098 | Used as a food source on SM/5 agar plates |
| Recombinant DNA reagent | pDXA-GFP2 (plasmid) | Dictybase | AF269235 | *Dictyostelium* expression vector with GFP |
| Recombinant DNA reagent | *adgf*^OE; AX4/*adgf*^OE | This paper | — | Full-length *adgf* (1.7 kb) cloned into pDXA-GFP2, and transformed into *adgf*- and WT-AX4 cells |
| Commercial assay or kit | ADA activity assay kit | Abcam | ab204695 | Used for adenosine deaminase activity |
| Commercial assay or kit | Adenosine quantification kit | Abcam | ab211094 | Used for total adenosine measurement |
| Commercial assay or kit | Ammonia assay kit | Sigma-Aldrich | AA0100 | Used for total ammonia estimation |
| Commercial assay or kit | cAMP-XP assay kit | Cell Signaling Technology | 4339 | For cAMP quantification |
| Software, algorithm | MEGA X | *Kumar et al., 2018* | RRID:SCR_000667 | Used for phylogenetic tree construction |
| Software, algorithm | SMART | *Letunic and Bork, 2018* | RRID:SCR_005026 | For domain analysis |
| Software, algorithm | BLAST | *Altschul et al., 1990* | RRID:SCR_004870 | For sequence similarity search |
| Software, algorithm | HADDOCK 2.4 | *Honorato et al., 2024* | RRID:SCR_014902 | For protein–protein docking |
| Software, algorithm | AlphaFold | *Jumper et al., 2021* | RRID:SCR_023662 | For tertiary structure prediction |
| Software, algorithm | PyMOL | Schrödinger, LLC | RRID:SCR_000305 | For protein visualization |
| Software, algorithm | ImageJ | NIH | RRID:SCR_003070 | For image analysis |
| Software, algorithm | GraphPad Prism | GraphPad Software | RRID:SCR_002798 | For statistical analyses |
| Software, algorithm | NIS-Elements D | Nikon | RRID:SCR_000667 | For microscopy image processing |
| Other | Nikon Eclipse TE2000 microscope | Nikon, Japan | RRID:SCR_023161 | For fluorescence and live imaging |
| Other | FACS Discover S8 Image Sorter | BD Biosciences, USA | — | Used for sorting GFP-positive/negative cells |

## Bioinformatic analyses of ADGF

The genomic sequence and the protein sequence of ADGF were obtained either from dictybase (http://dictybase.org) or NCBI database (https://www.ncbi.nlm.nih.gov/). The ADA domain within the DdADGF was identified by SMART and BLAST (http://smart.embl-heidelberg.de/) (https://blast.ncbi.nlm.nih.gov/Blast.cgi) (*Altschul et al., 1990*; *Letunic and Bork, 2018*) analyses. A phylogenetic tree was generated by aligning multiple amino acid sequences of ADGF from several taxa. Neighbour Joining approach and the MUSCLE alignment tool of the MEGAX programme were used for constructing the tree (*Saitou and Nei, 1987*). The tertiary structure of ADGF was obtained using the online programme alphafold (https://alphafold.ebi.ac.uk/) (*Jumper et al., 2021*), and PyMol was used for viewing the images. The expression profiles of *ada* and *ada2* during mouse and human gastrula development were retrieved from the marionilab.cruk.cam.ac.uk/MouseGastrulation2018/ and human-gastrula.net/ databases (*Pijuan-Sala et al., 2019*; *Tyser et al., 2021*), respectively. Protein–protein docking was carried out using High Ambiguity Driven Protein-Protein Docking (HADDOCK) version 2.4 (*Honorato et al., 2021*; *Honorato et al., 2024*).

## Culture and development of *Dictyostelium discoideum*

The *D. discoideum* WT-AX4 strain (DBS0237637) was obtained from Dictybase (http://dictybase.org/). The mutant strains used in this study were obtained from the GWDI bank (https://remi-seq.org), in *Dictyostelium* stock centre, North Western University, USA (*Gruenheit et al., 2021*). Strain identity was authenticated by diagnostic PCR to verify loss of the wild-type allele. Mutant strains were maintained under antibiotic selection to ensure retention of the insertion cassette. WT-AX4 cells or the mutant *adgf* derived from AX4 were cultured in modified maltose-HL5 medium (Formedium, UK) with 100,000 U/l penicillin and 0.1 g/l streptomycin. Three independent mutants (GWDI_17_D_7, GWDI_47_C_1, GWDI_132_H_3; insertion in exon 2 in all three mutants) of the *adgf* gene (DDB_G0275179) were obtained. The mound defects were identical in all three, and the strain GWDI_47_C_1 alone was characterized further. The culture was raised as a monolayer in Petri plates or grown in an Erlenmeyer flask in shaking conditions at 150 rpm and 22°C, to a cell density of $2–4 \times 10^6$ cells/ml. All cells were maintained below 10 passages and routinely verified to be mycoplasma free. The cells were also grown on SM/5 agar plates supplemented with *K. pneumoniae* at 22°C (2 g/l glucose, 2 g/l protease peptone, 0.4 g/l yeast extract, 1 g/l $MgSO_4·H_2O$, 0.66 g/l $K_2HPO_4$, 2.225 g/l $KH_2PO_4$, 1% Bactoagar, pH 6.4). For developmental assays, freshly starved cells were washed twice with ice cold KK2 buffer (2.25 g $KH_2PO_4$ and 0.67 g $K_2HPO_4$ per litre, pH 6.4), and plated on 1% non-nutrient KK2 agar plates at a density of $5 \times 10^5$ cells/cm$^2$ (*Nassir et al., 2019*). Thereafter, the plates were incubated in dark conditions at 22°C for development.

## *Dictyostelium* genomic DNA isolation

WT and *adgf* $^-$ cells grown axenically were harvested, and the pellet was resuspended in 1 ml of lysis buffer (50 mM Tris-Cl, pH 8; 10 mM EDTA; 0.8% SDS). To this mixture, 200 µl of Nonidet P-40 (NP40), a non-ionic detergent, was added, vortexed and centrifuged at $12,000 \times g$ for 15 min at room temperature (RT). The resultant pellet was gently vortexed, resuspended in 500 µl of lysis solution containing 200 µg/ml Proteinase K, and incubated at 65°C for 30 min. 300 µl of phenol: chloroform was added to this suspension, centrifuged at $18,000 \times g$ for 10 min at RT, and the resultant aqueous phase was extracted carefully. An equal amount of chloroform was added and centrifuged at $18,000 \times g$ for 10 min at RT. Following this, 750 µl of pure ethanol was added to the suspension, which was then centrifuged at $12,000 \times g$ for 15 min at 4°C. Genomic DNA was precipitated by adding twice the volume of absolute ethanol and 1/10th the volume of 3 M sodium acetate. After a 10-min centrifugation at $15,000 \times g$, the pellet was cleaned using 70% ethanol and stored at 4°C. Electrophoresis was conducted in TAE buffer at 50 V using a Medox power pack system (India), and the integrity of DNA was confirmed on a 1% agarose gel.

## Validation of the *adgf* mutant

To validate the blasticidin (*bsr*) resistance cassette insertion in the *adgf* mutant, WT and *adgf* $^-$ genomic DNA were isolated, and a diagnostic PCR was performed using the gene and *bsr* insert specific primers in accordance with the guidelines provided in the GWDI website. A qRT-PCR was carried out

to confirm the absence of *adgf* expression in the mutant cells. The primers used for mutant validation are listed in *Supplementary file 1B*.

## Quantitative real-time PCR

Total RNA was extracted from WT and *adgf* ⁻ cells using Trizol reagent (Favorgen, USA) at specified intervals (every 4 hr from 0 to 24 hr). cDNA was synthesized from the total RNA using a PrimeScript 1st Strand cDNA Synthesis Kit (Takara, Japan). Random primers from the manufacturer were used to generate the cDNA from a template of 1 µg total RNA. qRT-PCR was performed using SYBR Green Master Mix (Thermo Scientific, USA) and 1 µl of cDNA. The expression levels of *adgf*, *acaA*, cAMP receptor A (*carA*), phosphodiesterases (*pdsA*, *regA*), 5′ nucleotidase (*5′nt*), extracellular matrix A (*ecmA*), extracellular matrix B (*ecmB*), prespore A (*pspA*), countin (*ctn*), and small aggregate (*smlA*) were quantified with a QuantStudio Flex 7 (Applied Biosystems, USA). The mitochondrial large RNA subunit (*rnlA*) served as an internal control. qRT-PCR data analysis was conducted according to the method described by *Schmittgen and Livak, 2008*. The primer sequences used for qRT-PCR are provided in *Supplementary file 1D*.

## Generation of *adgf* overexpression construct

The full-length 1.7 kb *adgf* sequence was PCR-amplified using ExTaq polymerase (Takara, Japan) with WT genomic DNA as the template. The amplified product was then ligated into the pDXA-GFP2 vector at the HindIII and KpnI restriction sites. Both *adgf* ⁻ and WT cells were electroporated with this vector, and G418-resistant clones (10 µg/ml) were isolated for further analysis. The expression of *adgf* was confirmed by semi-quantitative PCR. The corresponding primer sequences are listed in *Supplementary file 1C*.

## Transformation of *Dictyostelium discoideum*

WT and *adgf* ⁻ cells grown axenically were harvested, washed twice with ice-cold EP buffer (10 mM $KH_2PO_4$, 10 mM $K_2HPO_4$, 50 mM sucrose, pH 6.2), and resuspended in 100 µl of EP++ solution (10 $K_2HPO_4$ mM, 10 mM $KH_2PO_4$, 50 mM sucrose, 1 mM $MgSO_4$, 1 mM $NaHCO_3$, 1 mM ATP, 1 µM $CaCl_2$) containing 10 µg of plasmid vector (*Nassir et al., 2019*) in pre-cooled cuvettes (Bio-Rad, USA). The cells were electroporated using a BTX ECM830 electroporator (Harvard Apparatus, USA) at 300 V with 2-ms pulses and five square wave pulses at 5-s intervals. The cells were then transferred to a Petri dish with 10 ml of HL5 medium and incubated at 22°C. After 24 hr, G418 (10 µg/ml) was added to the medium, and resistant colonies were selected for further analysis.

## Preparation of CM

WT and *adgf* ⁻ cells grown in HL5 medium were collected at the mid-log (ML) phase, resuspended in KK2 buffer at a density of $1 \times 10^7$ cells/ml, and incubated at 22°C with shaking for 20 hr. The clarified supernatant obtained after centrifugation was used for the experiments.

## Assay for ADA activity

Total ADA activity from *Dictyostelium* was determined as per the protocol of the manufacturer (Abcam, USA; Cat No: ab204695). For sample preparation, $5 \times 10^5$ cells/cm² were seeded onto KK2 agar plates, and at the mound stage, ice-cold ADA assay buffer was flooded, then vigorously pipetted to disrupt the mound integrity. The cell homogenate was agitated on a rotary shaker at 4°C for 15 min and then centrifuged at 12,000 rpm for 10 min in a cold microfuge tube. The supernatant was subjected to the ADA assay. This ADA test relies on adenosine to inosine formation. The intermediate formed combines with the probe to produce uric acid, which is quantified at 293 nm. BCA (Bicinchonic acid) assay kit (Thermoscientific, USA) was used for determining the protein concentration. One unit of ADA activity is defined as the amount of enzyme that hydrolyses adenosine to yield 1 µmol of inosine per min under the assay conditions.

## Adenosine quantification

Total adenosine levels from *Dictyostelium* mounds were measured according to the manufacturer's protocol, using the adenosine assay kit (Abcam, USA; Cat No: ab211094). Cells grown in HL5 were collected, washed and plated on KK2 agar plates. The cell lysis buffer was added to the plates with

mounds and mixed thoroughly. 50 µl of the lysate was mixed with 2 U of ADA and was subjected to incubation for 15 min at RT. Adenosine quantification involves the use of ADA and after a series of enzymatic reactions, an intermediate is formed which reacts with the adenosine probe generating a fluorescent product. Using a spectrofluorometer (Perkin Elmer, USA; $\lambda$ Ex = 544 nm/ $\lambda$ Em = 590), the fluorescence intensity was measured, which is proportional to the concentration of adenosine. The adenosine levels were quantified using the adenosine standard curve.

## Quantification of ammonia

Ammonia assay kit (Sigma-Aldrich, USA; Cat No: AA0100) was used for estimating the total ammonia levels. WT and *adgf* $^-$ cells developed on KK2 agar plates were sealed with parafilm and incubated at 22°C. The mounds were collected using lysis solution, and the debris was removed by centrifugation at 10,000× *g* for 10 min. The supernatant was used for further analysis. For the ammonia assay, 1 ml of assay reagent was mixed thoroughly with either 100 µl of samples or standards, incubated for 5 min at RT, and the absorbance was measured at 340 nm. Then, 10 µl of L-glutamate dehydrogenase (GDH) solution was added to each cuvette, and after a 5-min incubation at 25°C, the absorbance was measured again at 340 nm using a spectrophotometer (Eppendorf, Germany). In the presence of GDH, ammonia combines with α-ketoglutaric acid and reduced NADPH to produce L-glutamate and oxidized $NADP^+$. The decrease in absorbance at 340 nm is proportional to the ammonia concentration. The ammonia standard curve was used to calculate the ammonia levels.

## Volatile ammonia generation

To generate ammonia, 1 ml of 1 N NaOH and 1 ml of $NH_4Cl$ (concentrations used 0.1, 1, and 10 mM) were mixed thoroughly (*Thadani et al., 1977*; *Feit et al., 1990*) and from this mix, 2 ml was aliquoted in the upper half of the Petri dish. The other half of the plate with the *adgf* $^-$ mounds on KK2 agar was inverted, sealed and incubated at 22°C. To determine whether WT mounds, physically separated from the mutants, could rescue the mound arrest, WT (1 × 10^6 cells/cm^2) and *adgf* $^-$ (5 × 10^5 cells/cm^2) cells were developed on KK2 agar plates on either side of compartmentalized and sealed Petri plates. *adgf* $^-$ cells developed on either side of the dish served as controls.

## Quantification of cAMP

Using the cAMP-XP test kit and following the manufacturer's instructions, total cAMP levels were determined from both the WT and the mutant (Cell Signaling, USA). WT and *adgf* $^-$ mounds were disrupted and collected in 1 ml ice cold KK2 buffer. After centrifuging the pellet, 100 µl of 1X lysis solution was added, and the mixture was incubated in ice for 10 min. Subsequently, 50 µl of lysate and 50 µl of HRP-conjugated cAMP solution were added to the test plates, which were then shaken horizontally at RT. After 3-hr incubation, the wells were emptied and washed three times with 200 µl of 1X wash buffer. 100 µl of stop solution was added to halt the reaction, and the absorbance was measured at 450 nm using a spectrophotometer (Bio-Rad, USA). The cAMP standard curve was used to determine the cAMP levels.

## Visualization of cAMP waves using dark field optics

5 × 10^5 cells/cm^2 were developed on KK2 agar plates under moist, dark conditions to monitor the propagation of cAMP waves. Using a Nikon DS-5MC camera mounted on a Nikon SMZ-1000 stereo zoom and Nikon Eclipse TE2000 inverted microscope, a timelapse video of the aggregation was captured in real time (Nikon, Japan). cAMP optical density waves were displayed by subtracting the image pairs (*Siegert and Weijer, 1995*) after processing the images with the NIS-Elements D program or ImageJ (NIH, USA).

## Under agarose cAMP chemotaxis assay

The under-agarose cAMP chemotaxis assay (*Woznica and Knecht, 2006*; *Singh and Insall, 2022*) was carried out with cells obtained from WT and *adgf* $^-$ mounds. Briefly, the cells were starved in KK2 buffer at 1 × 10^7 cells/ml density. Three parallel troughs, each measuring 2 mm in width, were set up on a Petri dish containing 1% agarose. In the outer two troughs, 100 µl of WT and *adgf* $^-$ cell suspension was aliquoted, respectively, and 10 µM cAMP was added to the central trough. Cell migration towards cAMP was recorded every 30 s over a total duration of 20 min using the NIS-Elements D

software and an inverted Nikon Eclipse TE2000 microscope (Nikon, Japan). 35 cells were tracked each time and subsequently analysed using ImageJ (NIH, USA).

## Mixing WT with mutant cells

WT and *adgf⁻* cells cultured in HL5 medium were harvested at the ML phase by centrifugation, rinsed twice with ice-cold KK2 buffer. The cells mixed in different ratios (1:9, 2:8, and 1:1) were adjusted to a final density of $5 \times 10^5$ cells/cm², plated on 1% KK2 agar plates, and incubated at 22°C for development.

## Tracking cell fate after mixing

*Dictyostelium* cells harvested from fresh HL5 media were resuspended at a density of $1 \times 10^6$ cells/ml in KK2 buffer, followed by incubation at 22°C for 1.5 hr in shaking conditions with 0.2 μM DIL (Invitrogen, USA). DIL has been used extensively as a cell tracker in *C. elegans*, *Xenopus and* mice (*Schultz and Gumienny, 2012*; *Xu et al., 2020*; *Erdogan et al., 2016*). WT or *adgf⁻* cells labelled with DIL were mixed with the unlabelled cells in a ratio of 1:9, 2:8, and 1:1 and plated for development. In the ratios mentioned, the smaller fraction represents the stained cells. DIL is a lipophilic dye that selectively stains the plasma membranes of living cells, allowing visualization and analysis of cell morphology.

## NR staining of mounds and slugs

WT and *adgf⁻* cells harvested from HL5 cultures were resuspended at a density of $1 \times 10^7$ cells per ml in KK2 buffer and treated with 0.005% NR solution for 15 min at RT in shaking conditions (*Bonner, 1952*). The stained cells were washed twice with KK2 buffer, plated on buffered agar plates and thereafter, NR stained slugs were observed using an upright microscope.

## Cell-type-specific expression analysis of *adgf*

To determine the differential expression of *adgf*, if any, between the two major cell types, psp-GFP cells in AX4 background were developed on KK2 agar plates at a density of $5 \times 10^5$ cells/cm² until they reached the slug stage, as described by *Ratner and Borth, 1983*. Fluorescence was initially monitored using microscopy to confirm GFP expression. Slugs were then dissociated in 20 mM phosphate buffer containing 40 mM EDTA, pH 7.0 (*Nadin et al., 2000*), filtered through a 10-μm nylon mesh, chilled on ice, and using FACS (FACS Discover S8 Image Sorter (BD Biosciences, USA)) the two cell types were sorted. Sorting was performed at a rate of 10,000 cells/s with a 488-nm laser. GFP-positive prespore (psp) cells and GFP-negative prestalk (pst) cells were collected separately on ice over several hours. Total RNA was extracted from both cell populations, and the expression of *adgf* was analysed using RT-PCR.

## Cell–cell adhesion assay

After 14 h of development on phosphate buffered agar plates, WT and *adgf⁻* mounds were disaggregated by repeated pipetting and vortexing using ice-cold KK2 buffer. Dissociated cells were resuspended in KK2 buffer and incubated at 150 rpm for 45 min. Single and non-adherent cells were counted using a Neubauer chamber (*Lam et al., 1981*).

## Treatment of *Dictyostelium* cells with different compounds

A highly specific ADA inhibitor, DCF that inhibits both extra and intra-cellular ADAs (*Cha et al., 1975*) was mixed with the WT cell suspension and plated on 1% non-nutrient KK2 agar plates at a density of $5 \times 10^5$ cells/cm². Of the different DCF concentrations tested (10 nM, 50 nM, 100 nM, and 1 μM), 100 nM showed complete tip inhibition.

For the enzymatic rescue assay, 10 U of bovine ADA dissolved in KK2 buffer (Sigma-Aldrich, USA) was added onto mutant mounds. The concentrations of 8-Br-cAMP (2 mM), cAMP (0.1 mM, 0.5 mM), c-di-GMP (0.1 mM, 0.5 mM), adenosine (1 μM, 10 μM, 100 μM, 1 mM), caffeine (10 nM, 100 nM, 1 μM), and IBMX (0.5 mM) were based on previous publications (*Chen et al., 2017*; *Chen and Schaap, 2012*;

*Nassir et al., 2019*; *Siegert and Weijer, 1989*) and these compounds were added independently either on top of the mounds or supplemented while plating. All the fine chemicals were from Sigma-Aldrich, USA, except when mentioned.

## Microscopy

Microscopy was performed using a Nikon SMZ-1000 stereo zoom microscope with epifluorescence optics, Nikon 80i Eclipse upright microscope, or a Nikon Eclipse TE2000 inverted microscope, connected with a digital sight DS-5MC camera (Nikon). Image processing was carried out using NIS-Elements D (Nikon) or ImageJ.

## Statistical tools

Data analyses were carried out using Microsoft Excel (2016). Statistical significance was determined by paired or unpaired, two-tailed Student's *t*-test and ANOVA analysis (GraphPad Prism, version 7).

## Materials availability statement

Materials developed in this study are available upon request from the corresponding author.

## Acknowledgements

We gratefully acknowledge the help of NBRP Nenkin, Japan and *Dictyostelium* Stock Center (DSC), Northwestern University, USA for providing various strains and plasmids used in this study. We thank Prof. Richard Gomer, Texas A&M University, USA for providing *aprA* mutant strain. We thank Prof. Kei Inouye, Kyoto University, Japan, Prof. Kees Weijer, University of Dundee, UK, and Prof. Gad Shaulsky, Baylor College of Medicine, USA for their comments on an earlier version of the manuscript/ thesis. We acknowledge Dr. Srividhya, Facility-in-Charge, Flow Cytometry facility, Bio-SAIF, IIT Madras, for assistance with FACS and technical support. We gratefully acknowledge the help from Indian Council of Medical Research (ICMR), New Delhi for their support in this project. We thank Steve Hawthorne from Scholarly Memory for his feedback and suggestions on an earlier version of the manuscript.

## Additional information

### Funding

| Funder | Grant reference number | Author |
| --- | --- | --- |
| Indian Council of Medical Research | 53/18/2012-CMB/BMS | Baskar Ramamurthy |

The funders had no role in study design, data collection, and interpretation, or the decision to submit the work for publication.

### Author contributions

Pavani Hathi, Conceptualization, Formal analysis, Investigation, Writing – original draft; Baskar Ramamurthy, Conceptualization, Formal analysis, Supervision, Writing – review and editing

### Author ORCIDs

Pavani Hathi 
Baskar Ramamurthy 

Reviewer #1 (Public review): https://doi.org/10.7554/eLife.104855.4.sa1
Reviewer #2 (Public review): https://doi.org/10.7554/eLife.104855.4.sa2
Author response https://doi.org/10.7554/eLife.104855.4.sa3

## Additional files

### Supplementary files
MDAR checklist

Supplementary file 1. Compilation of supplementary tables (A–D) used in this study. Table A: Various deaminases annotated in *D. discoideum*. Table B: Primers used for adgf semi-quantitative PCR. Primers used for adgf overexpression and vector construction. Table D: Primers used for real-time PCR.

### Data availability
All data supporting the findings of this study are included in the article and its supplementary files. All numerical data associated with the figures have been deposited in Dryad.

The following dataset was generated:

| Author(s) | Year | Dataset title | Dataset URL | Database and Identifier |
|---|---|---|---|---|
| Hathi P, Ramamurthy B | 2025 | Extracellular adenosine deamination primes tip organizer development in *Dictyostelium* | https://doi.org/10.5061/dryad.76hdr7t8w | Dryad Digital Repository, 10.5061/dryad.76hdr7t8w |

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
