## [Editor Report · eLife Assessment]

During the development of the unicellular eukaryote Dictyostelium discoideum, cells aggregate into mounds, forming protrusions or tips, which then become the front of migrating slugs and the top of fruiting bodies. This **valuable** study identifies adenosine deaminase-related growth factor (ADGF) as a key regulator of tip formation and **convincingly** shows that ADGF catalyses the conversion of adenosine to ammonia, allowing ammonia to initiate tip formation, and then elucidates pathways upstream and downstream of ADGF. The authors discuss the intriguing possibility that mammalian ADGF may also similarly regulate development.

---

## [Referee Report · Reviewer #1 (Public review)]

Summary:

This work shows that a specific adenosine deaminase protein in Dictyostelium generates the ammonia that is required for tip formation during Dictyostelium development. Cells with an insertion in the adgf gene aggregate but do not form tips. A remarkable result, shown by several different ways, is that the adgf mutant can be rescued by exposing the mutant to ammonia gas. The authors also describe other phenotypes of the adgf mutant such as increased mound size, altered cAMP signaling, and abnormal cell type differentiation. It appears that the adgf mutant has defects the expression of a large number of genes, resulting in not only the tip defect but also the mound size, cAMP signaling, and differentiation phenotypes.

Strengths:

The data and statistics are excellent.

Comments on previous version:

Looks better, but I think you answered my questions (listed as weaknesses in the public review) in the reply to the reviewer but not in the paper. I'd suggest carefully thinking about my questions and addressing them in the Discussion (The authors have now done this).

---

## [Referee Report · Reviewer #2 (Public review)]

Summary:

The paper describes new insights into the role of adenosine deaminase-related growth factor (adgf), an enzyme that catalyses the breakdown of adenosine into ammonia and inosine, in tip formation during Dictyostelium development. The adgf null mutant has a pre-tip mound arrest phenotype, which can be rescued by external addition of ammonia. Analysis suggests that the phenotype involves changes in cAMP signaling possibly involving a histidine kinase dhkD, but details remain to be resolved.

Strengths:

The generation of an adgf mutant showed a strong mound arrest phenotype and successful rescue by external ammonia. Characterisation of significant changes in cAMP signaling components, suggesting low cAMP signaling in the mutant and identification of the histidine kinase dhkD as a possible component of the transduction pathway. Identification of a change in cell-type differentiation towards prestalk fate

Comments on previous version:

The revised version of the paper has improved significantly in terms of structure and clarity. The additional data on rescue of total cAMP production by ammonia (Fig. 7C) in the adgf- mutant and the 5-fold increased prespore expression of adgf RNA compared to prestalk cells (Fig 9) are useful data additions.

The link between changes in cAMP signaling (lower aca expression) and wave geometry (concentric waves rather than spiral waves) remains speculative.

I noted that Fig 6 contains different images than the previous version (Fig 7).

The statement "Interestingly, Klebsiella pneumoniae physically separated from the Dictyostelium adgf mutants in a partitioned dish, also rescues the mound arrest phenotype suggesting a cross-kingdom interaction that drives development" in the summary is rather overdone. All experiments were performed with axenic strains (no bacteria).

as is the sentence "Remarkably, in higher vertebrates, adgf expression is elevated during gastrulation and thus adenosine deamination may be a conserved process driving organizer development in different organisms"

The data supporting this in the supplementary information is hardly legible and poorly presented. What is shown is ADA expression in different tissues, not at different stages. I would suggest taking these figures out and concentrating the summary on the key mechanistic findings of the paper. (The authors have now done this.)

---

## [Author Response]

The following is the authors’ response to the previous reviews.

**Reviewer #1 (Public review):**
Summary:This work shows that a specific adenosine deaminase protein in Dictyostelium generates the ammonia that is required for tip formation during Dictyostelium development. Cells with an insertion in the ADGF gene aggregate but do not form tips. A remarkable result, shown in several different ways, is that the ADGF mutant can be rescued by exposing the mutant to ammonia gas. The authors also describe other phenotypes of the ADGF mutant such as increased mound size, altered cAMP signalling, and abnormal cell type differentiation. It appears that the ADGF mutant has defects in the expression of a large number of genes, resulting in not only the tip defect but also the mound size, cAMP signalling, and differentiation phenotypes.Strengths:The data and statistics are excellent.(1) Weaknesses: The key weakness is understanding why the cells bother to use a diffusible gas like ammonia as a signal to form a tip and continue development.

Ammonia can come from a variety of sources both within and outside the cells and this can be from dead cells also. Ammonia by increasing cAMP levels, trigger collective cell movement thereby establishing a tip in Dictyostelium. A gaseous signal can act over long distances in a short time and for instance ammonia promotes synchronous development in a colony of yeast cells (Palkova et al., 1997; Palkova and Forstova, 2000). The slug tip is known to release ammonia probably favouring synchronized development of the entire colony of Dictyostelium. However, after the tips are established ammonia exerts negative chemotaxis probably helping the slugs to move away from each other ensuring equal spacing of the fruiting bodies (Feit and Sollitto, 1987).

It is well known that ammonia serves as a signalling molecule influencing both multicellular organization and differentiation in Dictyostelium (Francis, 1964; Bonner et al., 1989; Bradbury and Gross, 1989). Ammonia by raising the pH of the intracellular acidic vesicles of prestalk cells (Poole and Ohkuma, 1981; Gross et al, 1983), and the cytoplasm, is known to increase the speed of chemotaxing amoebae (Siegert and Weijer, 1989; Van Duijn and Inouye, 1991), inducing collective cell movement (Bonner et al., 1988, 1989), favoring tipped mound development.

Ammonia produced in millimolar concentrations during tip formation (Schindler and Sussman, 1977) could ward off other predators in soil. For instance, ammonia released by Streptomyces symbionts of leaf-cutting ants is known to inhibit fungal pathogens (Dhodary and Spiteller, 2021). Additionally, ammonia may be recycled back into amino acids, as observed during breast cancer proliferation (Spinelli et al., 2017). Such a process may also occur in starving Dictyostelium cells, supporting survival and differentiation. These findings suggest that ammonia acts as both a local and long-range regulatory signal, integrating environmental and cellular cues to coordinate multicellular development.

(2) The rescue of the mutant by adding ammonia gas to the entire culture indicates that ammonia conveys no positional information within the mound.

Ammonia reinforces or maintains the positional information by elevating cAMP levels, favoring prespore differentiation (Bradbury and Gross, 1989; Riley and Barclay, 1990; Hopper et al., 1993). Ammonia is known to influence rapid patterning of Dictyostelium cells confined in a restricted environment (Sawai et al., 2002). In adgf mutants that have low ammonia levels, both neutral red staining (a marker for prestalk and ALCs) (Figure. S3) and the prestalk marker ecmA/ ecmB expression (Figure. 7D) are higher than the WT and the mound arrest phenotype can be reversed by exposing the adgf mutant mounds to ammonia.

Prestalk cells are enriched in acidic vesicles, and ammonia, by raising the pH of these vesicles and the cytoplasm (Davies et al 1993; Van Duijn and Inouye 1991), plays an active role in collective cell movement during tip formation (Bonner et al., 1989).

(3) By the time the cells have formed a mound, the cells have been starving for several hours, and desperately need to form a fruiting body to disperse some of themselves as spores, and thus need to form a tip no matter what.

Exposure of adgf mounds to ammonia, led to tip development within 4 h (Figure. 5). In contrast, adgf controls remained at the mound stage for at least 30 h. This demonstrates that starvation alone is not the trigger for tip development and ammonia promotes the transition from mound to tipped mound formation.

Many mound arrest mutants are blocked in development and do not proceed to form fruiting bodies (Carrin et al., 1994). Further, not all the mound arrest mutants tested in this study were rescued by ADA enzyme (Figure. S4A), and they continue to stay as mounds.

(4) One can envision that the local ammonia concentration is possibly informing the mound that some minimal number of cells are present (assuming that the ammonia concentration is proportional to the number of cells), but probably even a minuscule fruiting body would be preferable to the cells compared to a mound. This latter idea could be easily explored by examining the fate of the ADGF cells in the mound - do they all form spores? Do some form spores?Or perhaps the ADGF is secreted by only one cell type, and the resulting ammonia tells the mound that for some reason that cell type is not present in the mound, allowing some of the cells to transdifferentiate into the needed cell type. Thus, elucidating if all or some cells produce ADGF would greatly strengthen this puzzling story.

A fraction of adgf mounds form bulkier spore heads by the end of 36 h as shown in Figure. 2H. This late recovery may be due to the expression of other ADA isoforms. Mixing WT and adgf mutant cell lines results in a chimeric slug with mutants occupying the prestalk region (Figure. 8) and suggests that WT ADGF favours prespore differentiation. However, it is not clear if ADGF is secreted by a particular cell type, as adenosine can be produced by both cell types, and the activity of three other intracellular ADAs may vary between the cell types. To address whether adgf expression is cell type-specific, prestalk and prespore cells will be separated by fluorescence activated cell sorter (FACS), and thereafter, adgf expression will be examined in each population.

**Reviewer #2 (Public review):**
Summary:The paper describes new insights into the role of adenosine deaminase-related growth factor (ADGF), an enzyme that catalyses the breakdown of adenosine into ammonia and inosine, in tip formation during Dictyostelium development. The ADGF null mutant has a pre-tip mound arrest phenotype, which can be rescued by the external addition of ammonia. Analysis suggests that the phenotype involves changes in cAMP signalling possibly involving a histidine kinase dhkD, but details remain to be resolved.Strengths:The generation of an ADGF mutant showed a strong mound arrest phenotype and successful rescue by external ammonia. Characterization of significant changes in cAMP signalling components, suggesting low cAMP signalling in the mutant and identification of the histidine kinase dhkD as a possible component of the transduction pathway. Identification of a change in cell type differentiation towards prestalk fate(1) Weaknesses: Lack of details on the developmental time course of ADGF activity and cell type type-specific differences in ADGF expression.

adgf expression was examined at 0, 8, 12, and 16 h (Figure. 1), and the total ADA activity was assayed at 12 and 16 h (Figure. 3). Previously, the 12 h data was not included, and it’s been added now (Figure. 3A). The adgf expression was found to be highest at 16 h and hence, the ADA assay was carried out at that time point. Since the ADA assay will also report the activity of other three isoforms, it will not exclusively reflect ADGF activity.

Mixing WT and adgf mutant cell lines results in a chimeric slug with mutants occupying the prestalk region (Figure. 8) suggesting that WT adgf favours prespore differentiation. To address whether adgf expression is cell type-specific, prestalk and prespore cells will be separated by fluorescence activated cell sorter (FACS), and thereafter, adgf expression will be examined in each population.

(2) The absence of measurements to show that ammonia addition to the null mutant can rescue the proposed defects in cAMP signalling.

The adgf mutant in comparison to WT has diminished acaA expression (Fig. 6B) and reduced cAMP levels (Fig. 6A) both at 12 and 16 h of development. The cAMP levels were measured at 8 h and 12 h in the mutant.

We would like to add that ammonia is known to increase cAMP levels (Riley and Barclay, 1990; Feit et al., 2001) in Dictyostelium. Exposure to ammonia increases acaA expression in WT (Figure. 7B) and is likely to increase acaA expression/ cAMP levels in the mutant also (Riley and Barclay, 1990; Feit et al., 2001) thereby rescuing the defects in cAMP signalling. Based on the comments, cAMP levels will also be measured in the mutant after the rescue with ammonia.

(3) No direct measurements in the dhkD mutant to show that it acts upstream of adgf in the control of changes in cAMP signalling and tip formation.

cAMP levels will be quantified in the dhkD mutant after treatment with ammonia. The histidine kinases dhkD and dhkC are reported to modulate phosphodiesterase RegA activity, thereby maintaining cAMP levels (Singleton et al., 1998; Singleton and Xiong, 2013). By activating RegA, dhkD ensures proper cAMP distribution within the mound, which is essential for the patterning of prestalk and prespore cells, as well as for tip formation (Singleton and Xiong, 2013). Therefore, ammonia exposure to dhkD mutants is likely to regulate cAMP signalling and thereby tip formation.

**Reviewer #1 (Recommendations for the authors):**
(1) Lines: 47,48 - "The gradient of these morphogens along the slug axis determines the cell fate, either as prestalk (pst) or as prespore (psp) cells." - many workers have shown that this is not true - intrinsic factors such as cell cycle phase drive cell fate.

Thank you for pointing this out. We have removed the line and rephrased as “Based on cell cycle phases, there exists a dichotomy of cell types, that biases cell fate as prestalk or prespore (Weeks and Weijer, 1994; Jang and Gomer, 2011).

(2) Line 48 - PKA - please explain acronyms at first use.

Corrected

(3) Line 56 - The relationship between adenosine deaminase and ADGF is a bit unclear, please clarify this more.

Adenosine deaminase (ADA) is intracellular, whereas adenosine deaminase related growth factor (ADGF) is an extracellular ADA and has a growth factor activity (Li and Aksoy, 2000; Iijima et al., 2008).

(4) Figure 1 - where are these primers, and the bsr cassette, located with respect to the coding region start and stop sites?

The primer sequences are mentioned in the supplementary table S2. The figure legend is updated to provide a detailed description.

(5) Line 104 - 37.47% may be too many significant figures.

Corrected

(6) Line 123 - 1.003 Å may be too many significant figures.

Corrected

(7) Line 128 - Since the data are in the figure, you don't need to give the numbers, also too many significant figures.

Corrected

(8) Figure 3G - did the DCF also increase mound size? It sort of looks like it did.

Yes, the addition of DCF increases the mound size (now Figure. 2G).

(9) Figure 3I - the spore mass shown here for ADGF - looks like there are 3 stalks protruding from it; this can happen if a plate is handled roughly and the spore masses bang into each other and then merge

Thank you for pointing this out. The figure 3I (now Figure. 2I) is replaced.

(10) Lines 160-162 - since the data are in the figure, you don't need to give the numbers, also too many significant figures.

Corrected.

(11) Line 165 - ' ... that are involved in adenosine formation' needs a reference.

Reference is included.

(12) Line 205 - 'Addition of ADA to the CM of the mutant in one compartment.' - might clarify that the mutant is the ADGF mutant

Yes, revised to 'Addition of ADA to the CM of the adgf mutant in one compartment.'

(13) Lines 222-223 need a reference for caffeine acting as an adenosine antagonist.

Reference is included.

(14) Figure 8B - left - use a 0-4 or so scale so the bars are more visible.

Thank you for the suggestion. The scale of the y-axis is adjusted to 0-4 in Figure. 7B to enhance the visibility of the bars.

**Reviewer #2 (Recommendations for the authors):**
The paper describes new insights into the role of ADGF, an enzyme that catalyses the breakdown of adenosine in ammonia and inosine, in tip formation in Dictyostelium development.A knockout of the gene results in a tipless mound stage arrest and the mounds formed are somewhat larger in size. Synergy experiments show that the effect of the mutation is non-cell autonomous and further experiments show that the mound arrest phenotype can be rescued by the provision of ammonia vapour. These observations are well documented. Furthermore, the paper contains a wide variety of experiments attempting to place the observed effects in known signalling pathways. It is suggested that ADGF may function downstream of DhkD, a histidine kinase previously implicated in ammonia signalling. Ammonia has long been described to affect different aspects, including differentiation of slug and culmination stages of Dictyostelium development, possibly through modulating cAMP signalling, but the exact mechanisms of action have not yet been resolved. The experiments reported here to resolve the mechanistic basis of the mutant phenotype need focusing and further work.(1) The paper needs streamlining and editing to concentrate on the main findings and implications.

The manuscript will be revised extensively.

Below is a list of some more specific comments and suggestions.

(2) Introduction: Focus on what is relevant to understanding tip formation and the role of nucleotide metabolism and ammonia (see https://doi.org/10.1016/j.gde.2016.05.014) leading. This could lead to the rationale for investigating ADGF.

The manuscript will be revised extensively

(3) Lines 36-38 are not relevant. Lines 55-63 need shortening and to focus on ADGF, cellular localization, and substrate specificity.

The manuscript will be revised accordingly. Lines 36-38 will be removed, and the lines 55-63 will be shortened.

In humans, two isoforms of ADA are known including ADA1 and ADA2, and the Dictyostelium homolog of ADA2 is adenosine deaminase-related growth factor (ADGF). Unlike ADA that is intracellular, ADGF is extracellular and also has a growth factor activity (Li and Aksoy, 2000; Iijima et al., 2008). Loss-of-function mutations in ada2 are linked to lymphopenia, severe combined immunodeficiency (SCID) (Gaspar, 2010), and vascular inflammation due to accumulation of toxic metabolites like dATP (Notarangelo, 2016; Zhou et al., 2014).

(4) Results: This section would benefit from better streamlining by a separation of results that provide more mechanistic insight from more peripheral observations.

The manuscript will be revised and the peripheral observations (Figure. 2) will be shifted to the supplementary information.

(5) Line 84 needs to start with a description of the goal, to produce a knockout.

Details on the knockout will be elaborated in the revised manuscript. Line number 84 (now 75). Dictyostelium cell lines carrying mutations in the gene adgf were obtained from the genome wide Dictyostelium insertion (GWDI) bank and were subjected to further analysis to know the role of adgf during Dictyostelium development.

(6) Knockout data (Figure 1) can be simplified and combined with a description of the expression profile and phenotype Figure 3 F, G, and Figure 5. Higher magnification and better resolution photographs of the mutants would be desirable.

Thank you, as suggested the data will be simplified (section E will be removed) and combined with a description of the expression profile and, the phenotype images of Figure 3 F, G, and Figure 5 (now Figure. 2 F, G, and Figure. 4) will be replaced with better images/ resolution.

(7) It would also be relevant to know which cells actually express ADGF during development, using in-situ hybridisation or promoter-reporter constructs.

To address whether adgf expression is cell type-specific, prestalk and prespore cells will be separated by fluorescence activated cell sorter (FACS), and thereafter, adgf expression will be examined in each population.

(8) Figure 2 - Information is less directly relevant to the topic of the paper and can be omitted (or possibly in Supplementary Materials).

Figure. 2 will be moved to supplementary materials.

(9) Figures 4A, B - It is shown that as could be expected ada activity is somewhat reduced and adenosine levels are slightly elevated. However, the fact that ada levels are low at 16hrs could just imply that differentiation of the ADGF- cells is blocked/delayed at an earlier time point. To interpret these data, it would be necessary to see an ada activity and adenosine time course comparison of wt and mutant, or to see that expression is regulated in a celltype specific manner that could explain this (see above). It would be good to combine this with the observation that ammonia levels are lower in the ADGF- mutant than wildtype and that the mutant phenotype, mound arrest can be rescued by an external supply of ammonia (Figure 6).

In Dictyostelium four isoforms of ADA including ADGF are present, and thus the time course of total ADA activity will also report the function of other isoforms. Further, a number of pathways, generate adenosine (Dunwiddie et al., 1997; Boison and Yegutkin, 2019). ADGF expression was examined at 0, 8, 12 and 16 h (Fig 1) and the ADA activity was assayed at 12 h, the time point where the expression gradually increases and reaches a peak at 16 h. Earlier, we have not shown the 12 h activity data which will be included in the revised version. ADGF expression was found to be highly elevated at 16 h and adenosine/ammonia levels were measured at the two points indicated in the mutant.

(10) Panel 4C could be combined with other measurements trying to arrive at more insight in the mechanisms by which ammonia controls tip formation.

Panel 4C (now 3C) illustrates the genes involved in the conversion of cAMP to adenosine. Since Figure. 3 focuses on adenosine levels and ADA activity in both WT and adgf mutants, we have retained Panel 3C in Figure. 3, for its relevance to the experiment.

(11) There is a large variety of experiments attempting to link the mutant phenotype and its rescue by ammonia to cAMP signalling, however, the data do not yet provide a clear answer.

It is well known that ammonia increases cAMP levels (Riley and Barclay, 1990; Feit et al., 2001) and adenylate cyclase activity (Cotter et al., 1999) in D. discoideum, and exposure to ammonia increases acaA expression (Fig 7B) suggesting that ammonia regulates cAMP signaling. To address the concerns, cAMP levels will be quantified in the mutant after ammonia treatment.

(12) The mutant is shown to have lower cAMP levels at the mound stage which ties in with low levels of acaA expression (Figures 7A and B), also various phosphodiesterases, the extracellular phosphodiesterase pdsa and the intracellular phosphodiesterase regA show increased expression. Suggesting a functional role for cAMP signalling is that the addition of di cGMP, a known activator of acaA, can also rescue the mound phenotype (Figure 7E). There appears to be a partial rescue of the mound arrest phenotype level by the addition of 8Br-cAMP (fig 7C), suggesting that intracellular cAMP levels rather than extracellular cAMP signalling can rescue some of the defects in the ADGF- mutant. Better images and a time course would be helpful.

The relevant images will be replaced and a developmental time course after 8-Br-cAMP treatment will be included in the revised manuscript (Figure. 6D).

(13) There is also the somewhat surprising observation that low levels of caffeine, an inhibitor of acaA activation also rescues the phenotype (Figure 7F).

With respect to caffeine action on cAMP levels, the reports are contradictory. Caffeine has been reported to increase adenylate cyclase expression thereby increasing cAMP levels (Hagmann, 1986) whereas Alvarez-Curto et al., (2007) found that caffeine reduced intracellular cAMP levels in Dictyostelium. Caffeine, although is a known inhibitor of ACA, is also known to inhibit PDEs (Nehlig et al., 1992; Rosenfeld et al., 2014). Therefore, if caffeine differentially affects ADA and PDE activity, it may potentially counterbalance the effects and rescue the phenotype.

(14) The data attempting to asses cAMP wave propagation in mounds (Fig 7H) are of low quality and inconclusive in the absence of further analysis. It remains unresolved how this links to the rescue of the ADGF- phenotype by ammonia. There are no experiments that measure any of the effects in the mutant stimulated with ammonia or di-cGMP.

The relevant images will be replaced (now Figure. 6H). Ammonia by increasing acaA expression (Figure. 7B), and cAMP levels (Figure. 7C) may restore spiral wave propagation, thereby rescuing the mutant.

(15) A possible way forward could also come from the observation that ammonia can rescue the wobbling mound arrest phenotype from the histidine kinase mutant dhkD null mutant, which has regA as its direct target, linking ammonia and cAMP signalling. This is in line with other work that had suggested that another histidine kinase, dhkC transduces an ammonia signal sensor to regA activation. A dhkC null mutant was reported to have a rapid development phenotype and skip slug migration (Dev. Biol. (1998) 203, 345). There is no direct evidence to show that dhkD acts upstream of ADGF and changes in cAMP signalling, for instance, measurements of changes in ADA activity in the mutant.

cAMP levels will be quantified in the dhkD mutant after ammonia treatment and accordingly, the results will be revised.

(16) The paper makes several further observations on the mutant. After 16 hrs of development the adgf- mutant shows increased expression of the prestalk cell markers ecmA and ecmB and reduced expression of the prespore marker pspA. In synergy experiments with a majority of wildtype, these cells will sort to the tip of the forming slug, showing that the differentiation defect is cell autonomous (Fig 9). This is interesting but needs further work to obtain more mechanistic insight into why a mutant with a strong tip/stalk differentiation tendency fails to make a tip. Here again, knowing which cells express ADGF would be helpful.

The adgf mutant shows increased prestalk marker expression in the mound but do not form a tip. It is well known that several mound arrest mutants form differentiated cells but are blocked in development with no tips (Carrin et al., 1994). This is addressed in the discussions (539). To address whether adgf expression is cell type-specific, prestalk and prespore cells will be separated by fluorescence activated cell sorter (FACS), and thereafter, adgf expression will be examined in each population.

(17) The observed large mound phenotype could as suggested possibly be explained by the low ctn, smlA, and high cadA and csA expression observed in the mutant (Figure 3). The expression of some of these genes (csA) is known to require extracellular cAMP signalling. The reported low level of acaA expression and high level of pdsA expression could suggest low levels of cAMP signalling, but there are no actual measurements of the dynamics of cAMP signalling in this mutant to confirm this.

The acaA expression was examined at 8 and 12 h (Figure. 6B) and cAMP levels were measured at 12 and 16 h in the adgf mutants (Figure. 6A). Both acaA expression and cAMP levels were reduced, suggesting that cells expressing adgf regulate acaA expression and cAMP levels. This regulation, in turn, is likely to influence cAMP signaling, collective cell movement within mounds, ultimately driving tip development. Exposure to ammonia led to increased acaA expression (Figure. 7B) in in WT. Based on the comments above, cAMP levels will be measured in the mutant before and after rescue with ammonia.

(18) Furthermore, it would be useful to quantify whether ammonia addition to the mutant reverses mound size and restores any of the gene expression defects observed.

Ammonia treatment soon after plating or six hours after plating, had no effect on the mound size (Figure. 5G).

(19) There are many experimental data in the supplementary data that appear less relevant and could be omitted Figure S1, S3, S4, S7, S8, S9, S10.

Figure S8, S9, S10 are omitted. We would like to retain the other figures

Figure S1 (now Figure. S2): It is widely believed that ammonia comes from protein (White and Sussman, 1961; Hames and Ashworth, 1974; Schindler and Sussman, 1977) and RNA (Walsh and Wright, 1978) catabolism. Figure. S2 shows no significant difference in protein and RNA levels between WT and adgf mutant strains, suggesting that adenosine deaminaserelated growth factor (ADGF) activity serves as a major source of ammonia and plays a crucial role in tip organizer development in Dictyostelium. Thus, it is important to retain this figure.

Figure S3 (now Figure. S4): The figure shows the treatment of various mound arrest mutants and multiple tip mutants with ADA enzyme and DCF, respectively, to investigate the pathway through which adgf functions. Additionally, it includes the rescue of the histidine kinase mutant dhkD with ammonia, indicating that dhkD acts upstream of adgf via ammonia signalling. Therefore, it is important to retain this figure.

Figure S4 (now Figure. S5): This figure represents the developmental phenotype of other deaminase mutants. Unlike adgf mutants, mutations in other deaminases do not result in complete mound arrest, despite some of these genes exhibiting strong expression during development. This underscores the critical role of adenosine deamination in tip formation. Therefore, let this figure be retained.

Figure S7 (now Figure. S8): Figure S8 presents the transcriptomic profile of ADGF during gastrulation and pre-gastrulation stages across different organisms, indicating that ADA/ADGF is consistently expressed during gastrulation in several vertebrates (Pijuan-Sala et al., 2019; Tyser et al., 2021). Notably, the process of gastrulation in higher organisms shares remarkable similarities with collective cell movement within the Dictyostelium mound (Weijer, 2009), suggesting a previously overlooked role of ammonia in organizer development. This implies that ADA may play a fundamental role in regulating morphogenesis across species, including Dictyostelium and vertebrates. Therefore, we would like to retain this figure.

(20). Given the current state of knowledge, speculation about the possible role of ADGF in organiser function in amniotes seems far-fetched. It is worth noting that the streak is not equivalent to the organiser. The discussion would benefit from limiting itself to the key results and implications.

The discussion is revised accordingly by removing the speculative role of ADGF in organizer function in amniotes. The lines “It is likely that ADA plays a conserved, fundamental role in regulating morphogenesis in Dictyostelium and other organisms including vertebrates” have been removed.